# Loss of PHF8 induces a viral mimicry response by activating endogenous retrotransposons

Yanan Liu [1,4], Longmiao Hu [1,4], Zhengzhen Wu [1,4], Kun Yuan[1], Guangliang Hong[2], Zhengke Lian [1], Juanjuan Feng [1], Na Li[1], Dali Li [1], Jiemin Wong [1], Jiekai Chen [3], Mingyao Liu [1], Jiangping He [2] ✉ & Xiufeng Pang [1] ✉

Immunotherapy has become established as major treatment modality for multiple types of solid tumors, including colorectal cancer. Identifying novel immunotherapeutic targets to enhance anti-tumor immunity and sensitize current immune checkpoint blockade (ICB) in colorectal cancer is needed. Here we report the histone demethylase PHD finger protein 8 (PHF8, KDM7B), a Jumonji C domain-containing protein that erases repressive histone methyl marks, as an essential mediator of immune escape. Ablation the function of PHF8 abrogates tumor growth, activates anti-tumor immune memory, and augments sensitivity to ICB therapy in mouse models of colorectal cancer. Strikingly, tumor PHF8 deletion stimulates a viral mimicry response in color-ectal cancer cells, where the depletion of key components of endogenous nucleic acid sensing diminishes PHF8 loss-mediated antiviral immune responses and anti-tumor effects in vivo. Mechanistically, PHF8 inhibition elicits H3K9me3-dependent retrotransposon activation by promoting pro-teasomal degradation of the H3K9 methyltransferase SETDB1 in a demethylase-independent manner. Moreover, PHF8 expression is anti-correlated with canonical immune signatures and antiviral immune responses in human colorectal adenocarcinoma. Overall, our study establishes PHF8 as an epigenetic checkpoint, and targeting PHF8 is a promising viral mimicry-inducing approach to enhance intrinsic anti-tumor immunity or to conquer immune resistance.

Colorectal cancer is one of the leading causes of cancer-related death worldwide[1]. Immunotherapy that disrupts immune evasion by targeting the immune checkpoint exerts an increasingly critical role in treating colorectal cancer in both resectable and non-resectable patients[2]. Nevertheless, tumors frequently evade immune surveillance. Most patients present primary or secondary resistance to immune checkpoint blockade (ICB) treatments through diverse mechanisms[3,4], and only a fraction of colorectal cancer patient show lasting responses. Therefore, identifying novel immunomodulatory targets to inflame tumors and enhance tumor intrinsic immunogenicity or combinations that potentiate current immunotherapeutic approaches is crucial.

[1]Shanghai Key Laboratory of Regulatory Biology and School of Life Sciences, East China Normal University, Shanghai, China. [2]Guangzhou Laboratory, Guangzhou, China. [3]Guangzhou Institutes of Biomedicine and Health, Chinese Academy of Sciences, Guangzhou, China. [4]These authors contributed equally: Yanan Liu, Longmiao Hu, Zhengzhen Wu. ✉e-mail: he_jiangping@gzlab.ac.cn; xfpang@bio.ecnu.edu.cn

Epigenetic regulators play critical roles in tumor initiation, progression, and even in therapy resistance. Recent evidence indicate that epigenetic factors are essential for tumor immune microenvironment remodeling[5]. For example, the DNA methyltransferase DNMT[6], the histone methyltransferase EZH2[7], the histone chaperone ASF1A[8], the H3K9 methyltransferase SETDB1[9] and the lysine-specific demethylases 1 A (LSD1)[10] and 5B (KDM5B)[11] have been reported to regulate anti-tumor immunity through different mechanisms. Targeting these epigenetic factors is considered as a promising approach to potentiate ICB treatments, opening a new avenue for cancer therapy. Recent work shows that epigenetic therapies converge with cancer immunotherapy through a mode of an antiviral response, named 'viral mimicry'. Viral mimicry responses are triggered by endogenous nucleic acids that are derived from aberrantly transcribed endogenous retrotransposons within the human genome[6,7,9–12]. The induction of viral mimicry by epigenetic regulation can shape antitumor immune responses and decrease cancer cell fitness[5]. However, epigenetic factors that regulate endogenous retrotransposons and further modulate anti-tumor immunity are still largely unknown.

Plant homeodomain finger protein 8 (PHF8, also known as KDM7B) is a histone lysine demethylase of the Jumonji C protein family. PHF8 binds to lysine methylated histone H3 lysine 4 (H3K4) through its N terminal PHD domain and erases repressive histone marks (H3K9me1, H3K9me2, H4K20me1, and H3K27me2) through its Jumonji C domain[13,14]. PHF8 is involved in several cellular and molecular processes, such as ribosomal RNA transcription[15], neuronal differentiation[16], genome stability[17], and cell cycle progression[14]. Prior studies have shown that mutations in PHF8 are associated with X-linked mental retardation and cleft lip/cleft palate[18]. Recent studies have revealed that PHF8 is aberrantly expressed in several human malignancies, such as gastric cancer[19], melanomas[20], breast cancer[21], colorectal cancer[22], prostate cancer[23], and acute lymphoblastic leukemia[24]. The underlying molecular mechanisms of PHF8 in cancer biology have been gradually discovered in specific tumor contexts. For example, PHF8 directly controls the TGF-β signaling pathway, thus leading to melanoma invasion and metastasis[20]. PHF8 primes the transcriptional activation of SNAI1, which contributes to epithelial to mesenchymal transition in breast cancer[25]. PHF8 interacts with c-Jun to modulate the PKCα-Src-PTEN axis in HER2-negative advanced gastric cancer[19]. However, the function and mechanism of tumor PHF8 in anti-tumor immunity have not yet been elucidated.

In this study, we identify that PHF8 functions as an immune suppressor that limits anti-tumor immunity in colorectal cancer. We discover that PHF8 silences endogenous retrotransposons and restrain cytosolic nucleic acid sensing. Accordingly, PHF8 loss activates endogenous retrotransposons, provokes antiviral immune responses and significantly enhances therapeutic efficacy of ICB treatments. Overall, our findings provide important evidence for harnessing epigenetic modulators such as PHF8 for cancer immunotherapy through a mechanism of viral mimicry responses.

## Results

### PHF8 loss induces a growth-inhibiting immune response
PHF8 plays a critical role in various developmental and disease processes[13]. However, its role in anti-tumor immunity is unclear. Recently, an in vivo epigenetic CRISPR screen has been conducted to identify cell-intrinsic epigenetic regulators of tumor immunity[8]. Phf8 was one of potential candidate genes in this screen, and targeting Phf8, when lost, possibly enhanced sensitivity to anti-PD-1 therapy. This prompted us to explore the role of PHF8 in tumor immune evasion. To examine the functional role of PHF8 in tumor growth in vitro, we deleted Phf8 in mouse models of colorectal adenocarcinoma (CT26 and MC38) using the CRISPR-Cas9 technology (Supplementary Fig. 1a). Knockout (KO) of Phf8 in CT26 or MC38 cells did not impair cell proliferation ability in vitro, as measured by colony-forming

(Supplementary Fig. 1b) and growth curve assays (Supplementary Fig. 1c). We then inoculated CT26 or MC38 cells into immunodeficient (Balb/c nude) mice for an in vivo study. There were no significant differences between Phf8 KO groups and the vector control group in terms of tumor growth (Fig. 1a, c) and the survival (Fig. 1b, d). To confirm these observations, we additionally set up xenograft mouse models of breast and pancreatic cancers using Phf8-deficient murine breast cancer cells (4T1) and pancreatic cancer cells (KPC). Our results consistently showed that knockout of Phf8 did not impair the growth of 4T1 and KPC xenografts in immunodeficient hosts (Supplementary Fig. 1d–g). Moreover, genetic knockout Phf8 resulted in comparable tumor progression in immunodeficient Rag2^-/- mice that do not have mature T and B cells (Supplementary Fig. 1h, i). These results suggest that PHF8 loss exerts little impacts on tumor cell proliferation in vitro and tumor growth in immunodeficient hosts in vivo.

Next, to determine the involvement of an intact immune system, we inoculated Phf8 KO and the vector control CT26 or MC38 cells into immunocompetent mice. Notably, in contrast to Phf8 wild-type counterparts, Phf8-KO tumor cells were completely rejected in syngeneic immunocompetent mice (Fig. 1e–h). Consistent results were observed in the 4T1 and KPC tumor models (Supplementary Fig. 1j–m). Moreover, Phf8-KO tumor cells had a marked growth disadvantage over wild-type cells in an in vivo competitive assay[26] (Supplementary Fig. 1n, o). These results indicate that anti-tumor efficacy of PHF8 deficiency is dependent on an intact immune system. Accordingly, ectopic expression of Phf8 accelerated tumor growth (Fig. 1i) and reduced survival (Fig. 1j) in immunocompetent mice. In summary, these results substantiate that PHF8 loss induces potent anti-tumor immunity.

### PHF8 loss induces anti-tumor immune memory and improves in vivo anti-tumor effects of PD-1 blockade
We observed that Phf8 knockout potently inhibited tumor formation in immunocompetent mice, and those mice could remain long-term tumor-free (Fig. 1e–h). This promoted us to speculate that PHF8 loss might induce an anti-tumor immune memory. In further experiments, we examined whether those tumor-free mice could resist a rechallenge with parental tumor cells. We injected lethal doses of wild-type tumor cells with intact Phf8 into mice that had already rejected Phf8-KO tumors. We found that mice that rejected CT26 Phf8-KO tumors (Fig. 1k, l) or MC38 Phf8-KO tumors (Supplementary Fig. 1p–r) were protected against rechallenge by their wild-type counterparts. Intriguingly, Phf8 wild-type 4T1 tumors grew much slower in rechallenged hosts inoculated with CT26 Phf8-KO tumors (Supplementary Fig. 1s–u), indicating an anti-tumor immune memory. Furthermore, we performed RNA-sequencing (RNA-seq) analysis (GSE212779) of shPhf8 and the vector control CT26 tumors (Supplementary Fig. 1v) and found that Phf8 expression was anti-correlated with canonical immune signatures, including T cell activation and adaptive immune response pathways (Supplementary Fig. 1w), implying that PHF8 maintained an immune-excluded phenotype. We further quantified immune effector cells using flow cytometry analysis. Our data consistently showed that CD8^+ T cells and activated (CD44^hi CD62L^lo) CD8^+ T cells were markedly increased in shPhf8 CT26 tumors as compared with the control tumors (Supplementary Fig. 1x, y). Moreover, the immunofluorescence density of granzyme B^+ (GZMB^+) and interferon-γ^+ (IFN-γ^+) CD8^+ T cells also significantly increased in Phf8-deficient CT26 tumors (Fig. 1m), implying that depletion of tumor PHF8 promotes adaptive anti-tumor CD8^+ T cell immune responses and results in tumor growth repression.

Our observations of enhanced tumor cell immunogenicity and increased CD8^+ T cell immune responses caused by Phf8 depletion suggest that Phf8-deficient tumors may be susceptible to ICB therapy. To evaluate whether PHF8 deficiency could potentiate ICB therapy, we used a PD-1 monoclonal antibody (anti-PD-1) to treat immunocompetent mice inoculated with the vector control cells or shPhf8 cells. As

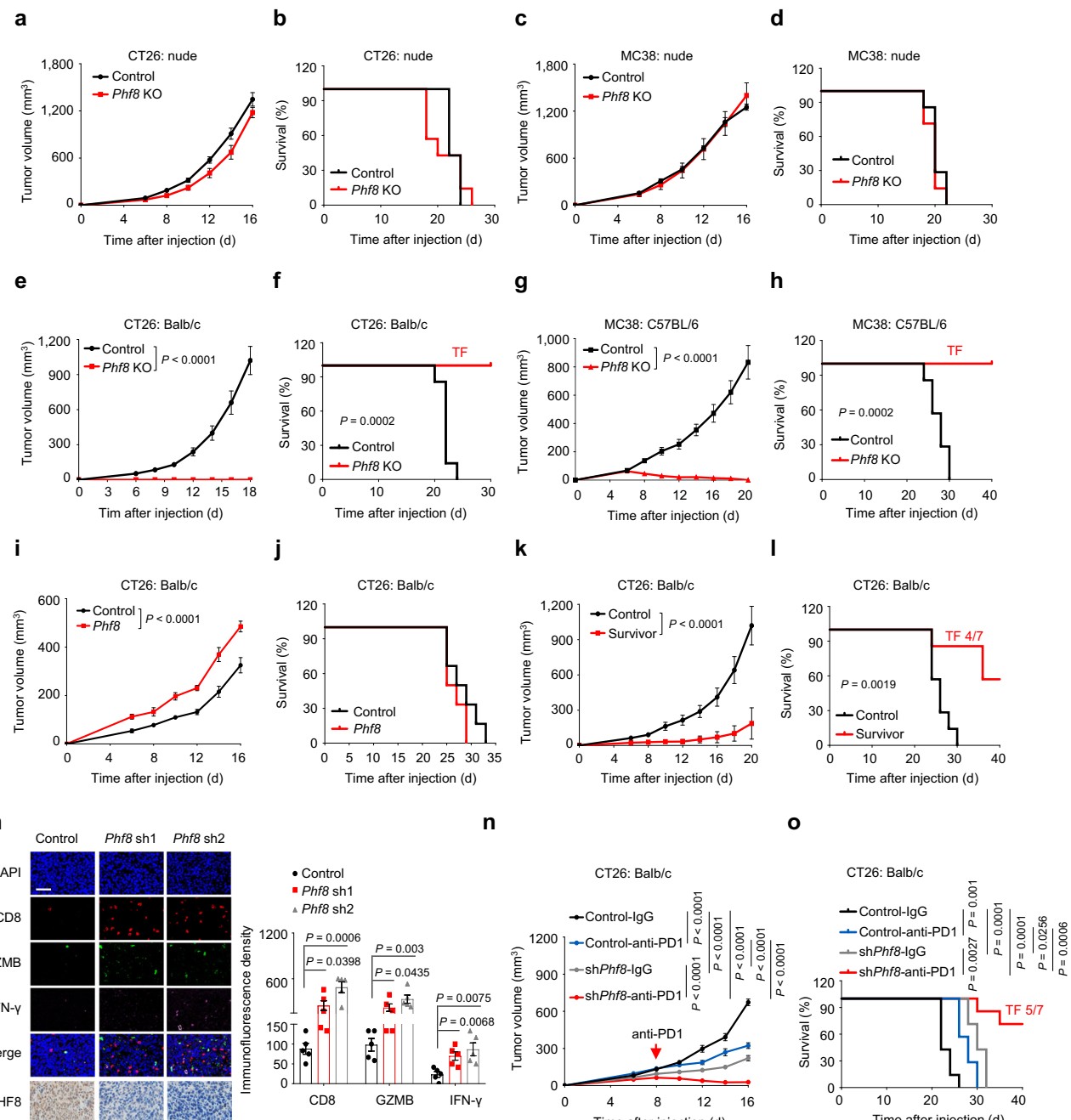

**Fig. 1 | PHF8 loss induces a growth-inhibiting immune response. a–d** Tumor growth curves **a**, **c** and Kaplan-Meier survival curves **b**, **d** of *Phf8* KO CT26 and MC38 cells and their corresponding vector control cells in immunodeficient mice. In each cases, about 500,000 (CT26) or 1,000,000 (MC38) tumor cells were inoculated subcutaneously into nude mice, and tumor formation was monitored. *n* = 7 tumors per group. **e–h** Tumor growth curves **e**, **g** and Kaplan-Meier survival curves **f**, **h** of *Phf8* KO CT26 and MC38 cells and their corresponding vector control cells in syngeneic immunocompetent mice. *n* = 7 tumors per group. **i**, **j** Tumor growth curves **i** and Kaplan-Meier survival curves **j** of Balb/c mice subcutaneously injected with about 500,000 the vector control or *Phf8* (exogenously transduced pLVX-*Phf8* vector) CT26 cells. *n* = 6 tumors per group. **k**, **l** tumor growth curves **k** and Kaplan-Meier survival curves **l** of host mice after rechallenge with 1,000,000 *Phf8* wild-type CT26 cells in Balb/c mice that remained tumor-free for 38 days after initial challenge with 500,000 *Phf8* KO CT26 cells. Control referred to tumor-naive Balb/c

mice challenged with *Phf8* wild-type CT26 cells. *n* = 7 tumors per group. **m** Immunofluorescence staining (*left*) and quantitative data (*right*) of CD8$^+$ (CD8$^+$ T cells), granzyme B$^+$ (GZMB$^+$), interferon-γ$^+$ (IFN-γ$^+$) in the vector control and sh*Phf8* CT26 tumors. *n* = 5 biologically independent samples. Fluorescent density for each sample was analyzed. Immunohistochemical images shows *Phf8* knock-down efficiency. Scale bar, 50 μm. **n**, **o** Tumor growth curves **n** and Kaplan-Meier survival curves **o** of Balb/c mice injected with 500,000 *Phf8* wild-type CT26 cells that were treated with IgG control or anti-PD-1 antibodies, or the same number of sh*Phf8* CT26 cells treated with IgG control or anti-PD-1 antibodies. IgG or anti-PD-1 treatment began at day 8. *n* = 7 tumors per group. Data are presented as the mean ± sem. TF, tumor free. Two-way ANOVA in **e**, **g**, **i**, **k**, and **n**, log-rank test in **f**, **h**, **l**, and **o**, and unpaired two-sided Student's *t*-test in **m**. Source data are provided as a Source Data file.

expected, we observed a reasonable therapeutic efficacy of anti-PD-1 in control mice bearing CT26 tumors. In sharp contrast, *Phf8* knockdown enabled anti-PD-1 to eradicate CT26 tumors (Fig. 1n, o). We additionally performed sensitizing experiments using the ICB-resistant 4T1 murine breast cancer model[27]. We found that co-treatment with anti-PD-1 and sh*Phf8* further decreased 4T1 tumor volume (Supplementary Fig. 1z, *left*) and prolonged mouse survival (Supplementary Fig. 1z, *right*) as compared with sh*Phf8* or anti-PD1 treatment alone. Taken together, these results suggest that tumor PHF8 is critical for intrinsic resistance to spontaneous and immunotherapy-induced tumor immunity.

## PHF8 abrogation triggers antiviral immune responses

To determine how PHF8 suppresses anti-tumor immunity, we performed transcriptomic analysis (GSE212779) of *Phf8* wild-type, KO and *Phf8* reconstitution (*Phf8* KO + *Phf8*) CT26 tumor cells. GO enrichment analysis demonstrated that terms related to 'interferon response' and 'antiviral response' were significantly upregulated in *Phf8* KO cells compared with the wild-type control, whereas *Phf8* reconstitution could specifically inverse all these pathways (Fig. 2a and Supplementary Data 1–4), suggesting a PHF8-dependent effect. As indicated by KEGG analysis, *Phf8* KO CT26 cells exhibited activation of multiple viral infection and antiviral host-defense pathways; however, *Phf8* reintroduction preferentially suppressed these responses (Fig. 2b, Supplementary Fig. 2a and Supplementary Data 5, 6). Gene set enrichment analysis (GSEA) further showed that pathways that induce interferon responses, including cytosolic DNA-sensing pathways and signaling pathways of RIG-I like receptor, Toll-like receptor, and NOD-like receptor, were activated by *Phf8* loss (Supplementary Fig. 2b, *upper*). Strikingly, introduction of wild-type *Phf8* into *Phf8* KO cells significantly repressed these events (Supplementary Fig. 2b, *lower*). Importantly, inflammation- and infection-related pathways and intracellular virus sensing signals were consistently activated in *Phf8* knockdown CT26 tumors from immunocompetent mice (Fig. 2c and Supplementary Fig. 2c). Collectively, these data suggest that *Phf8* depletion activates anti-tumor immunity by triggering interferon and antiviral responses.

As molecular events in PHF8-mediated anti-tumor immunity, type 1 interferons and interferon-stimulated genes (ISGs) were significantly upregulated after *Phf8* loss in CT26 and MC38 cells, whereas *Phf8* reintroduction could block this effect (Fig. 2d). *Phf8*-depedent modulation of MDA5, RIG-I, cGAS, phosphorylated-TBK1, -IRF3, and -STAT1 protein expression was further observed in both CT26 and MC38 cells (Fig. 2e). *Phf8* loss-mediated activation of interferon responses could be enhanced by the addition of IFN-γ (Fig. 2f, g), reinforcing the role of PHF8 in regulating interferon signaling. Notably, depletion of the RNA sensor MDA5 (encoded by *Ifih1*) or RIG-I (encoded by *Ddx58*), RNA adaptor MAVS, DNA sensor cGAS or DNA adaptor STING (encoded by *Sting1*) diminished ISGs induction in *Phf8* KO CT26 (Fig. 3a, b) and MC38 cells (Fig. 3c, d), suggesting that both the cytosolic RNA-sensing and DNA-sensing pathways were essential for interferon pathway activation induced by PHF8 loss. Consistent with this, simultaneous abrogation of MDA5 and cGAS, RIG-I and cGAS, or MAVS and STING impaired sh*Phf8*-induced anti-tumor effects in vivo (Fig. 3e–g). These results suggest that endogenous nucleic acid sensing pathways are essential for PHF8 depletion-induced interferon responses and tumor regression.

It is well established that MHC-I molecules responsible for antigen processing and presentation (APP) are downstream targets of the interferon pathway[28]. RNA-seq data showed that APP-associated genes were markedly upregulated in *Phf8* KO CT26 cells (Supplementary Fig. 3a). Real-time quantitative PCR (RT-qPCR) and fluorescence activated cell sorting analysis further confirmed the increased expression of APP-associated genes (Supplementary Fig. 3b, c) and cell surface expression of MHC-I molecules in *Phf8* KO CT26 and MC38 cells (Supplementary Fig. 3d). Taken together, our results establish that

inhibition of PHF8 led to activation of antiviral immune responses and APP pathway in colorectal cancer cells.

## PHF8 silences endogenous retrotransposons

Activated endogenous retrovirus (ERV) transcripts lead to the formation of intracellular dsRNA, which is recognized by pattern recognition receptors and subsequently triggers interferon responses[10]. We thus speculate that *Phf8* loss triggers endogenous nucleic acid sensing pathways through activating retrotransposons. To this end, we analyzed strand-specific RNA-seq data and found that the expression of a cluster of retrotransposons was regulated by tumor PHF8 (Fig. 4a, b, Supplementary Fig. 4a and Supplementary Data 7, 8), including long terminal repeat (LTR)-containing ERVs and non-LTR elements, such as long interspersed nuclear elements (LINEs) and short interspersed nuclear elements (SINEs) (Supplementary Fig. 4b, c). Notably, several repetitive elements, such as *RLTR46B*, *MER68B*, *LTR67B*, *RLTR13D3*, *MERV1_I-int*, and *L1Md_A*, could be significantly upregulated upon *Phf8* depletion (Log$_2$ FD > 1, $P < 0.05$) and significantly repressed by *Phf8* reintroduction (Log$_2$ FD < -1, $P < 0.05$), highlighting a critical role of PHF8 in tightly controlling retrotransposons (Fig. 4c). We further confirmed the altered expression of representative retrotransposons using RT-qPCR experiments (Fig. 4d). Knockdown of *RLTR31_Mur* or *RLTR46B* could independently decrease the expression of the key IFN-induced antiviral factor *Oasl2* (Fig. 4e), demonstrating a potential link between *Phf8*-regulated ERVs and ISGs. We noticed that these *Phf8*-regulated retrotransposons exhibited increased concurrent sense and antisense transcription after *Phf8* depletion and decreased bidirectional transcription upon *Phf8* reintroduction (Fig. 4f), raising the probability that the retrotransposon transcripts could pair and form cytosolic dsRNAs.

To determine directly whether there was an increase in dsRNA abundance in cells after *Phf8* loss, we performed fluorescence microscopy and flow cytometry analysis using the monoclonal J2 antibody that specifically recognized dsRNA[29]. Our results showed that dsRNA formation was significantly induced in *Phf8* KO cells (Fig. 4g, h) as well as in *Phf8*-deficient CT26 tumors (Supplementary Fig. 4d).

As *PHF8* loss also upregulated the cytosolic DNA-sensing pathway that contributed to interferon responses, we deduced that the released cytosolic DNA was probably generated through the reverse transcription of retrotransposons. This speculation was subsequently validated by the data that the reverse transcriptase inhibitor Lamivudine and Nevirapine were capable to repress *Phf8*-loss-triggered activation of cytosolic DNA abundance (Supplementary Fig. 4e), cytosolic DNA sensing components (Supplementary Fig. 4f) and ISG expression (Supplementary Fig. 4g). Taken together, these data demonstrate that cytosolic nucleic acids with retrotransposon origin contribute to antiviral immune responses in *Phf8*-deficient cells.

## PHF8 ablation activates H3K9me3-marked retrotransposons in a demethylase-independent manner

To assess whether PHF8 represses retrotransposons dependent on its enzymatic activity, we reintroduced wild-type *Phf8* or *Phf8*[H1247A] (expressing catalytically inactive PHF8)[14] into *Phf8* KO cells and found that they could reverse the de-repression of retrotransposons and immunostimulatory genes (Fig. 5a). This modulatory effect was further verified by the RT-qPCR data from knock-in cells expressing a H274A variant of PHF8 (Fig. 5b). These results suggest that demethylase activity of PHF8 was unnecessary for retrotransposon and IFN repression. Notably, this PHF8 catalytically inactive form possessed comparable tumor promoting ability with the wild-type PHF8 (Fig. 5c), further indicating that the role of PHF8 in tumor immune evasion was independent of its catalytic activity.

We next carried out the chromatin immunoprecipitation followed by sequencing (ChIP−seq) analysis (GSE212779) to comprehensively explore how PHF8 restrain retrotransposon expression. We found that

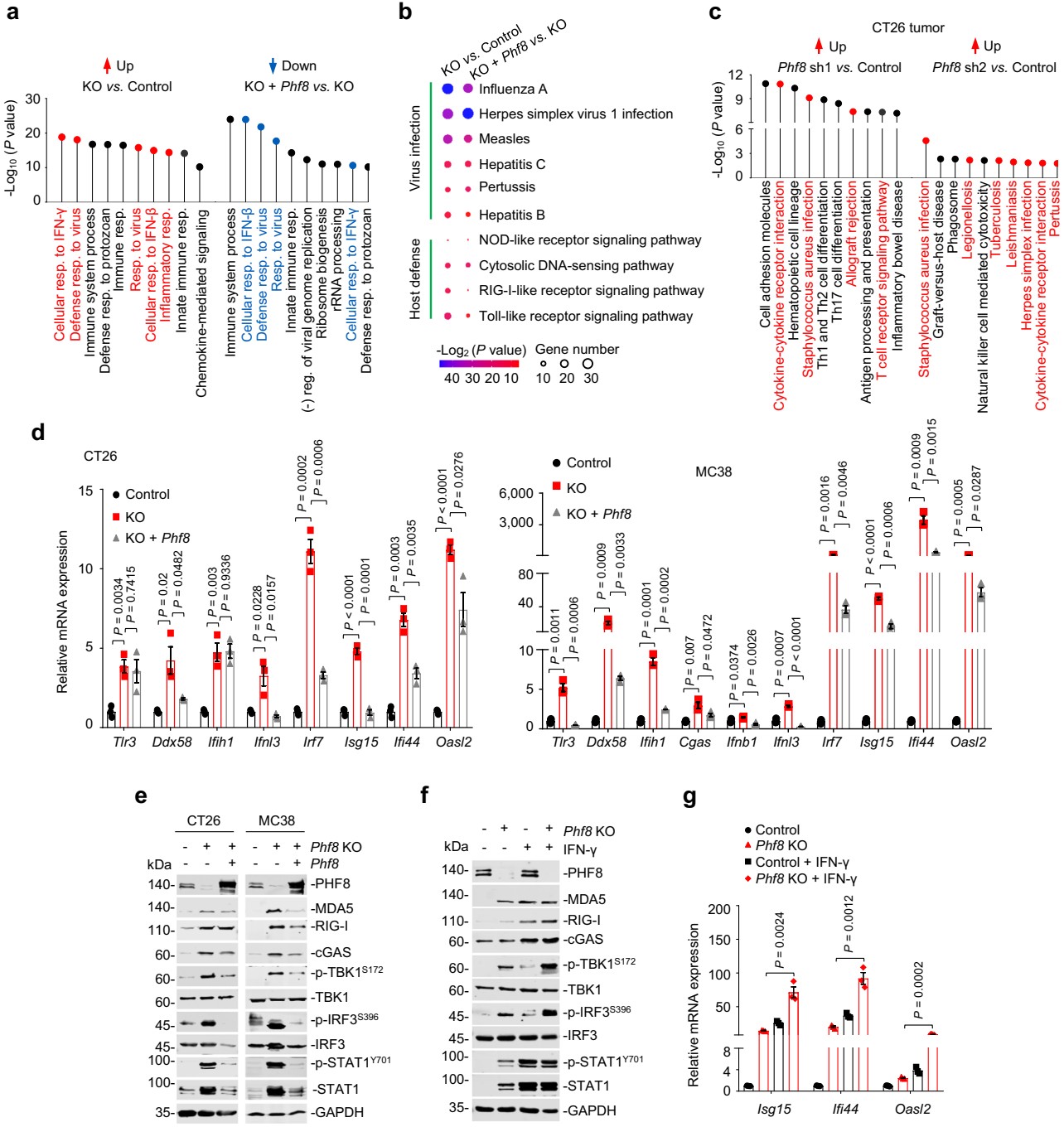

**Fig. 2 | Deletion of *PHF8* activates endogenous antiviral and interferon responses. a** Gene ontology (GO) analysis of RNA-seq data showing top 10 pathways that are upregulated in *Phf8* KO CT26 cells compared with the control cells (*left*) and top 10 pathways that are downregulated in *Phf8* KO + *Phf8* CT26 cells compared with *Phf8* KO cells (*right*). Graph displays category scores as −log₁₀ (*P* value) from Fisher's exact test. Red (upregulated GO terms) and blue (downregulated GO terms) indicates GO terms related to antiviral response or interferon (IFN) response. *n* = 3; (-), negative; resp., response; reg., regulation. **b** KEGG pathway analysis of RNA-seq data showing viral infection and host-defense responses of upregulated differentially expressed genes in *Phf8* KO CT26 cells compared with the control cells (*left*), and corresponding pathways of downregulated differentially expressed genes in *Phf8* KO + *Phf8* CT26 cells compared with *Phf8* KO cells (*right*), *n* = 3; graph displays category scores as −log₂ (*P* value) from Fisher's exact test and gene number. **c** KEGG pathway analysis of RNA-seq data showing upregulated differentially expressed genes in sh*Phf8* CT26 tumors compared with the control

tumors. Graph displays category scores as −log₁₀ (*P* value) from Fisher's exact test. Red indicates inflammation- and infection-related pathways and intracellular virus sensing signals. **d** Real-time quantitative PCR (RT-qPCR) analysis of transcripts of selected IFNs and ISGs in the control, *Phf8* KO and *Phf8* KO + *Phf8* murine tumor cells. **e** Western blot analysis of DNA and RNA sensors as well as interferon response related protein expression in the control, *Phf8* KO, and *Phf8* KO + *Phf8* murine tumor cells. **f** Western blot analysis of the expression of DNA and RNA sensors and downstream signaling proteins in the vector control and *Phf8* KO CT26 cells treated with or without 20 ng/mL IFN-γ for 24 hours. **g** RT-qPCR analysis of transcripts of selected ISGs in the control and *Phf8* KO CT26 cells treated with or without IFN-γ. For **d** and **g**, values are expressed as mean ± SEM. *n* = 3 biologically independent samples. Unpaired two-sided Student's *t*-test in **d** and **g**. The immunoblots in **e** and **f** are representative of three independent experiments. Source data are provided as a Source Data file.

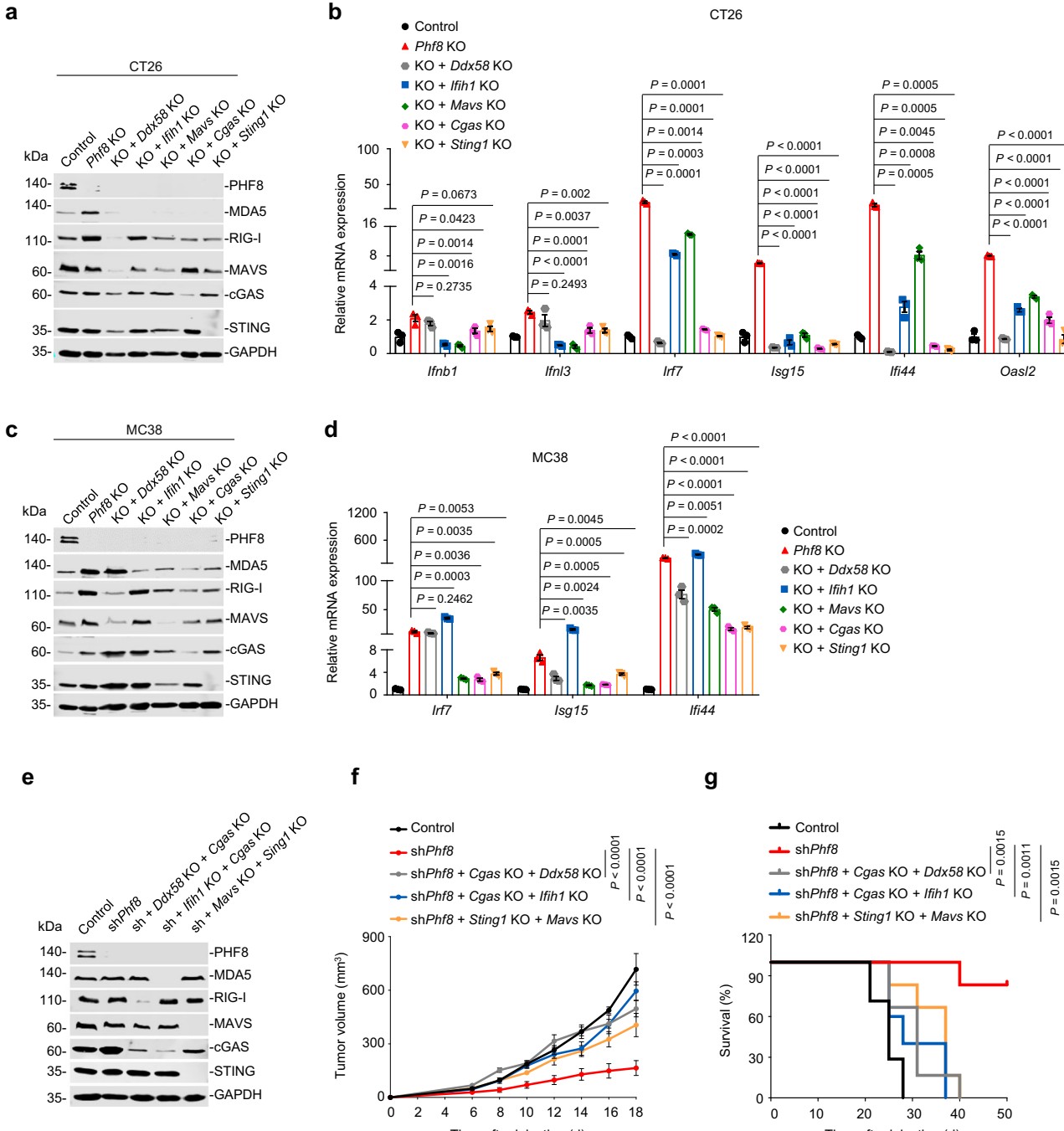

**Fig. 3 | Deletion of *PHF8* activates interferon responses and induces tumor regression through endogenous nucleic acid sensing pathways. a** Western blot analysis showing double knockout efficiency of *Phf8* plus *Ddx58*, *Ifih1*, *Mavs*, *Cgas* or *Sting1* in CT26 cells. **b** RT-qPCR analysis of CT26 *Phf8* KO cells double with *Ddx58* KO, *Ifih1* KO, *Mavs* KO, *Cgas* KO or *Sting1* KO. *ns*, not significant. **c** Western blot analysis showing double knockout efficiency of *Phf8* plus *Ddx58*, *Ifih1*, *Mavs*, *Cgas* or *Sting1* in MC38 cells. **d** RT-qPCR analysis of MC38 *Phf8* KO cells double with *Ddx58* KO, *Ifih1* KO, *Mavs* KO, *Cgas* KO or *Sting1* KO. **e** Immunoblot analysis showing the efficiency of *Phf8* knockdown plus double knockout of *Ddx58* and *Cgas*, *Ifih1* and

*Cgas*, or *Cgas* and *Sting1* in MC38 cells. **f, g** Tumor growth curves (**f**) and Kaplan-Meier survival curves (**g**) of C57BL/6 mice injected with MC38 cells of the indicated genotypes. The control (*n* = 7), sh*Phf8* (*n* = 6), sh*Phf8* + *Cgas* KO + *Ddx58* KO (*n* = 6), sh*Phf8* + *Cgas* KO + *Ifih1* KO (*n* = 5), and sh*Phf8* + *Sting1* KO + *Mavs* KO (*n* = 6). For **b** and **d**, values are expressed as mean ± SEM. *n* = 3 biologically independent samples. Unpaired two-sided Student's *t*-test in **b**, **d**, two-way ANOVA in **f** and log-rank test in **g**. The immunoblots in **a**, **c**, and **e** are representative of three independent experiments. Source data are provided as a Source Data file.

PHF8 binding signals were mainly enriched at active promoters and gene body regions rather than transposable elements in CT26 cells (Fig. 5d and Supplementary Fig. 5a). PHF8 has been reported to act on monomethylated and dimethylated H3 lysine 9 (H3K9me1/2), dimethylated H3 lysine 27 (H3K27me2), and monomethylated histone H4 lysine 20 (H4K20me1), and serves as a transcriptional activator[13,14].

ChIP-seq data (GSE211526) demonstrated that H3K9me1, H3K9me2 and H3K27me2 peaks were marginally affected by *Phf8* loss at the genome-wide level (Supplementary Fig. 5b). H4K20me1 signals (GSE211526) that are mostly enriched at gene promoters were decreased upon *Phf8* deficiency (Supplementary Fig. 5c), which concurred with previous reports that PHF8 positively regulates gene

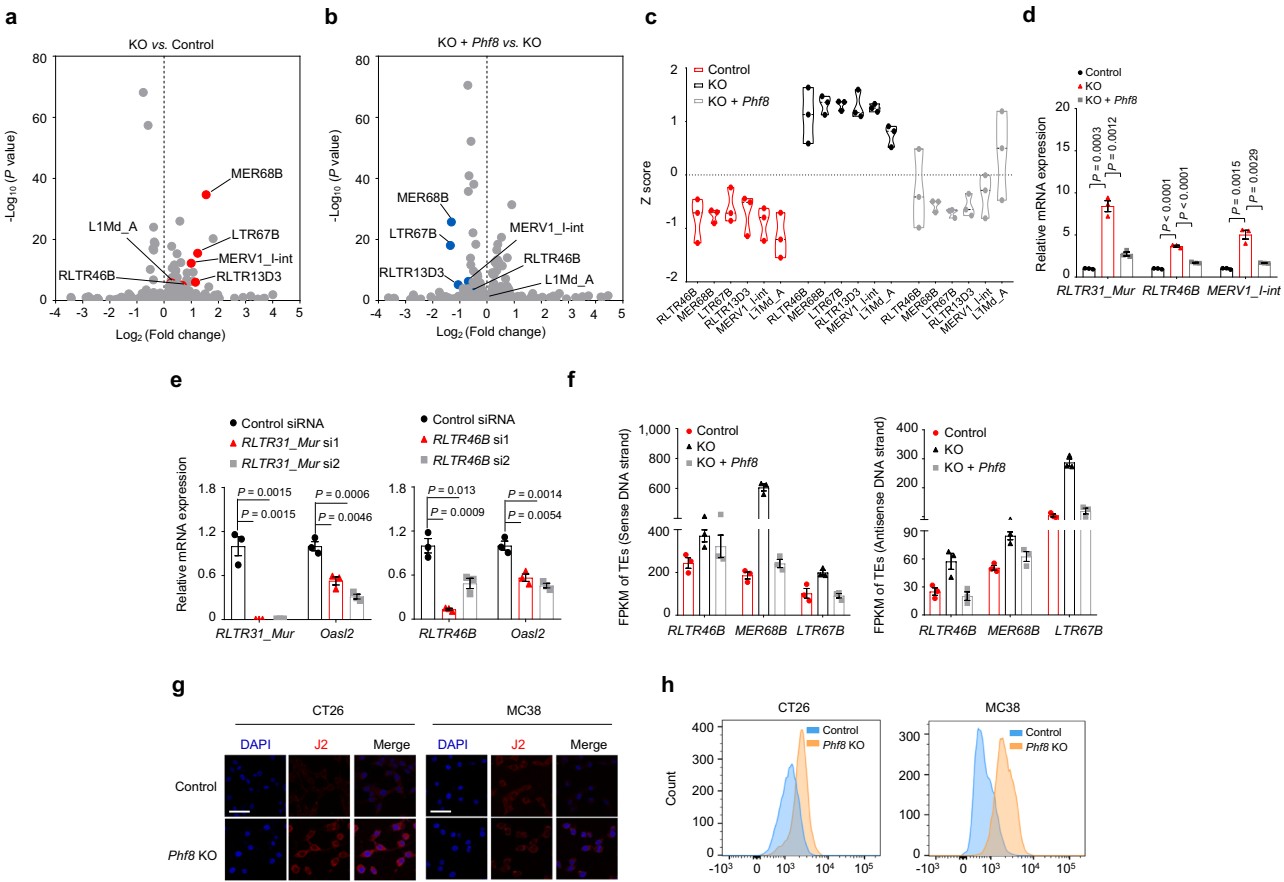

**Fig. 4 | PHF8 loss induces endogenous retrotransposon expression and dsRNA formation. a** Scatterplots of differentially expressed transposons in *Phf8* KO CT26 cells relative to the control according to strand-specific paired-end RNA-seq analysis. *P* value was calculated using DESeq2 package (v1.20.0) (see details in "Methods"). **b** Scatterplots of differentially expressed transposons in *Phf8* KO + *Phf8* CT26 cells relative to *Phf8* KO cells. *P* value was calculated using DESeq2 package (v1.20.0) (see details in "Methods"). **c** The expression levels of *RLTR46B*, *MER68B*, *LTR67B*, *RLTR13D3*, *MERV1_I-int* and *L1Md_A* in CT26 control, *Phf8* KO, *Phf8* KO + *Phf8* cells. *n* = 3. **d** RT-qPCR analysis of transcripts of retrotransposons in the vector control, *Phf8* KO and *Phf8* KO + *Phf8* CT26 cells. **e** RT-qPCR analysis of *Phf8* KO cells that were treated with the control, *RLTR31_Mur* (*left*) or *RLTR46B* (*right*) targeting siRNAs for 24 h. **f** Bar plots derived from strand-specific RNA-seq of the control, *Phf8* KO and *Phf8* KO + *Phf8* CT26 cells. Retrotransposons are shown as mRNA expression (FPKM) of reads derived from 5′or 3′transcribed products. *n* = 3, values are expressed as mean ± SEM. **g** Immunofluorescence staining of dsRNA with J2 antibody (red) and DAPI (blue) in CT26 (*left*) and MC38 cells (*right*). Representative images of 3 independent experiments. Scale bar, 50 μm. **h** Flow cytometry analysis using J2 antibody for dsRNA expression in CT26 (*left*) and MC38 cells (*right*). For **d**–**f**, values are expressed as mean ± SEM. *n* = 3 biologically independent samples. Unpaired two-sided Student's *t*-test in **d** and **e**. Source data are provided as a Source Data file.

expression with activity towards H4K20me1[14,20]. In agreement, our ChIP followed by quantitative PCR (ChIP–qPCR) results further showed that H3K9me1, H3K9me2, H3K27me2 and H4K20me1 levels at *Phf8*-regulated retrotransposons were slightly affected by *Phf8* depletion (Supplementary Fig. 5d).

Retrotransposon silencing is best illustrated during early mammalian development. Retrotransposons, including intracisternal A particle (IAP) retrotransposons and LINE1 elements, have been reported to be primarily controlled by repressive H3K9me3 heterochromatin[30]. Therefore, we extrapolated the functional importance of H3K9me3 enrichment to PHF8-regulated retrotransposons. Our ChIP–seq data demonstrated a marked decrease in H3K9me3 levels in *Phf8* KO cells compared with the control cells (Fig. 5e). H3K9me3-bound peaks primarily localized in transposable element regions (Fig. 5f) but declined in *Phf8* KO cells (Supplementary Fig. 5e). Specifically, the H3K9me3 levels of *RLTR31_Mur*, *RLTR46B*, *MERV1_I-int*, *L1Md_A*, *IAPEz-int*, *MMERVK10C-int*, and *IAPEY3-int* were significantly reduced upon *Phf8* ablation (Fig. 5g and Supplementary Fig. 5f). Decreased H3K9me3 binding to these retrotransposon loci was further verified by ChIP-qPCR analysis (Fig. 5h).

Results from transposase-accessible chromatin using sequencing (ATAC-seq) analysis (GSE211526) showed that PHF8 loss increased chromatin accessibility in a group of retrotransposon loci in cells with or without IFN-γ incubation (Fig. 5i). Retrotransposon that gained accessibility in *Phf8* KO CT26 cells were enriched for motifs recognized by TEAD, Jun-AP1, RUNX, NF-E2, and Bach (Supplementary Fig. 5g), which are crucial regulators in cancer progression, cancer-associated stress responses and genome topology[31,32]. In addition, ISGs, such as *Ifi44*, *Irf8* and *Stat5b*, gained increased accessibility after PHF8 loss (Supplementary Fig. 5h), highlighting a repressive role of PHF8 on retrotransposon-mediated interferon responses.

### PHF8 facilitates the nuclear stabilization of SETDB1 to silence H3K9me3-marked retrotransposons
We next investigated the mechanism by which PHF8 silences H3K9me3-established retrotransposons. H3K9me3 chromatin marks at retrotransposons are mainly regulated by the H3K9 methyltransferase SETDB1 or SUV39H1[33–35]. We first analyzed their mRNA expression according to our RNA–seq data (GSE212779) and observed little changes between *Phf8*-deficient cells (or xenograft tumors) and

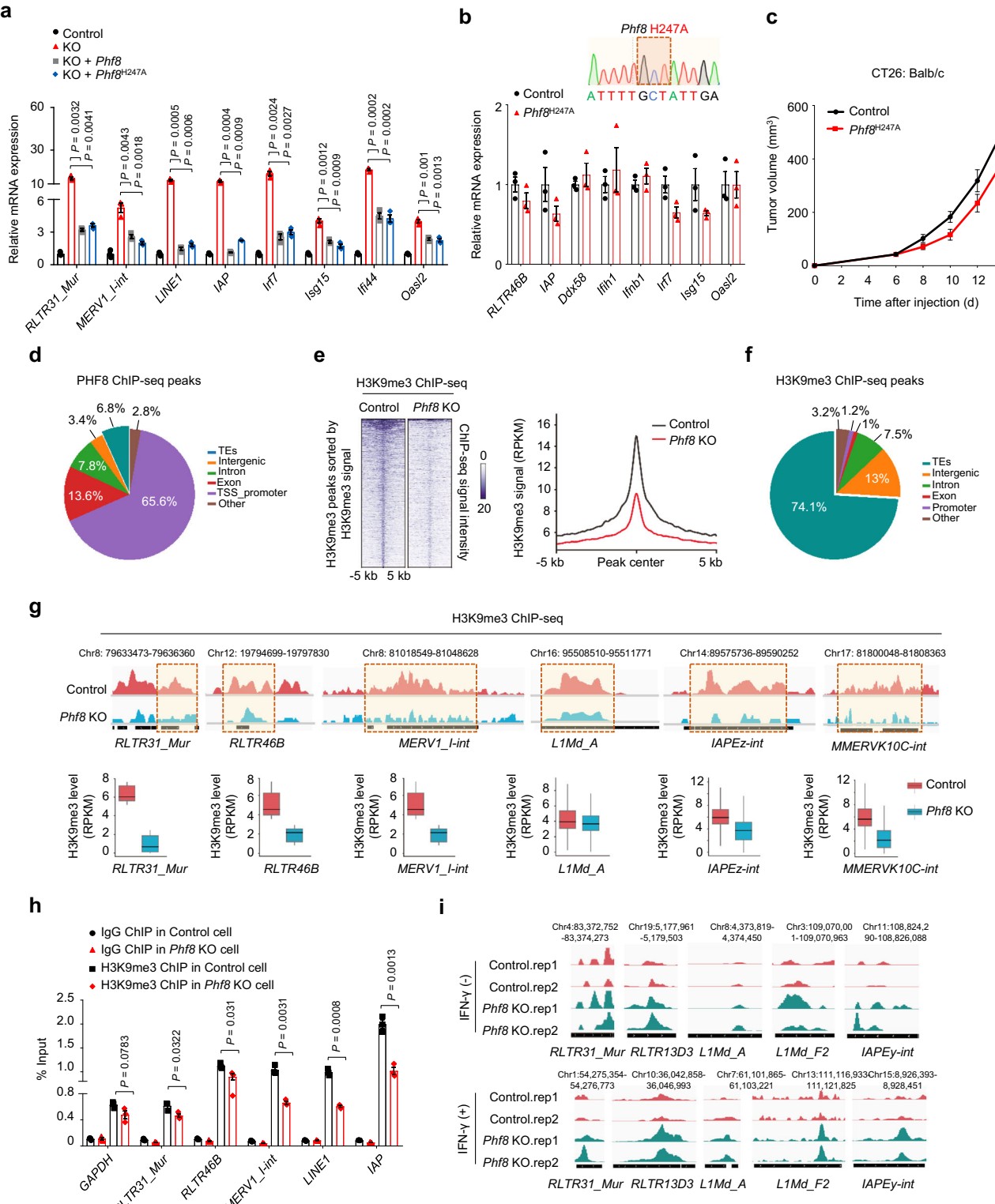

the control cells (Fig. 6a). Consistent with this, our RT-qPCR analysis further confirmed this bioinformatics results (Fig. 6b). In contrast to the almost unchanged protein levels of SUV39H1 in the cytoplasm and nucleus, nuclear SETDB1 abundance was strikingly decreased in the absence of PHF8 (Fig. 6c). Treatment of *Phf8* KO cells with the proteasome inhibitor MG132 resulted in complete recovery of SETDB1 expression in the nucleus of CT26 and MC38 cells (Fig. 6d, e), indicating that SETDB1 enters the nucleus but in doing so becomes a substrate for proteasomal degradation in the absence of PHF8.

SETDB1 plays a central role in repressive chromatin processes during genome evolution; however, the regulation of SETDB1 expression and stability remain largely unknown to date. To our best knowledge, ATF7IP (also known as MCAF1) is a well-recognized factor that functions as a SETDB1-interacting protein and specifically regulates SETDB1 abundance in the nucleus[36,37]. Therefore, we speculated that PHF8 may affect ATF7IP expression or disrupt the interaction of ATF7IP and SETDB1 to control nuclear SETDB1 abundance. To test this hypothesis, we examined ATF7IP expression and its interaction with

**Fig. 5 | PHF8 represses H3K9me3-marked retrotransposons. a** RT–qPCR analysis of transcripts of selected retrotransposons and ISGs in the control, *Phf8* KO, *Phf8* KO + *Phf8* and *Phf8* KO + *Phf8*[H247A] (catalytically inactive mutant) CT26 cells. **b** RT-qPCR analysis of transcripts of selected retrotransposons, IFNs and ISGs in the control and *Phf8*[H247A] knock-in CT26 cells. Sanger sequencing results are shown for the *Phf8* locus from the *Phf8*[H247A] knock-in CT26 cells. **c** Tumor growth curves of *Phf8*[H247A] knock-in (*n* = 8) and the corresponding control CT26 cells (*n* = 8) in syngeneic immunocompetent mice. **d** Genomic annotations of PHF8 binding peaks in CT26 cells. **e** Heatmap of the H3K9me3 ChIP–seq signals within H3K9me3 peaks in the control and *Phf8* KO CT26 cells (*left*). Read-count tag density pileups of H3K9me3 profiles on H3K9me3 peaks (*right*). **f** Genomic annotations of H3K9me3 binding peaks in CT26 cells. **g** Integrative genomic viewer (IGV) screenshots of

aggregated H3K9me3 ChIP–seq signals of selected transposable element (TE) loci (*upper*). Average change of H3K9me3 levels for corresponding TE loci (*lower*). Two independent samples per group. Boxplots denote the medians and the inter-quartile ranges (IQR). The whiskers of a boxplot are the lowest datum still within 1.5 IQR of the lower quartile and the highest datum still within 1.5 IQR of the upper quartile. **h** ChIP–qPCR showing binding patterns of H3K9me3 on retrotransposons. **i** IGV screenshots of aggregated ATAC–seq signals of selected TE loci in the control and *Phf8* KO CT26 cells treated with or without 20 ng/mL IFN-γ. For **a**, **b**, and **h**, values are expressed as mean ± SEM. *n* = 3 biologically independent samples. Unpaired two-sided Student's *t*-test in **a** and **h**. Source data are provided as a Source Data file.

SETDB1 in *Phf8* vector control and KO cells. Our results unexpectedly demonstrated that PHF8 loss showed little impacts on neither ATF7IP mRNA expression (Supplementary Fig. 6a) nor protein abundance in the nucleus (Supplementary Fig. 6b). Moreover, the interaction of SETDB1 and ATF7IP in the nucleus was not affected by PHF8 ablation either (Supplementary Fig. 6c). These results suggest that PHF8-mediated SETDB1 stabilization was not associated with ATF7IP.

To further explore the underlying mechanism by which PHF8 regulated SETDB1 stability, we conducted co-immunoprecipitation experiments and found that PHF8 could directly interact with SETDB1 in both CT26 and MC38 cells (Fig. 6f). In attempt to clarify the relationship of PHF8 and SETDB1 in regulating genome-wide H3K9me3 levels, we further compared the distribution of H3K9me3 in *Phf8* KO and *Setdb1* KO cells. We collected publicly available H3K9me3 ChIP-seq data (GSE155972) in *Setdb1* KO murine tumor cells[9] and discovered 30936 *Setdb1*-dependent H3K9me3 peaks (Supplementary Fig. 6d), approximately 30% of which could be reduced by *Phf8* depletion (Supplementary Fig. 6d, *upper right*). When noted, H3K9me3 peaks that were co-regulated by SETDB1 and PHF8 mainly located in transposable element regions (Supplementary Fig. 6e, *upper*).

Our data, along with others[14,20,38], have shown that PHF8 barely binds to transposable element regions but mainly binds to gene promoter regions (Fig. 5d). Our analysis further showed that PHF8 did not bind to neither PHF8-SETDB1 co-regulated transposons nor SETDB1-regulated transposons (Supplementary Fig. 6f). Based on these findings, we next investigated whether PHF8 regulates the recruitment of SETDB1 to retrotransposons. We carried out ChIP-qPCR assays and found that SETDB1 recruitment at selected retrotransposons was significantly impaired in *Phf8* KO cells (Fig. 6g). Loss of *Setdb1* led to a significant induction of retrotransposons and ISGs (Fig. 6h), whereas ectopic expression of SETDB1 repressed these effects in *Phf8*-deficient cells (Fig. 6i, j). In summary, our findings suggest that PHF8 facilitates SETDB1 stabilization to establish H3K9me3 marks on heterochromatin (Supplementary Fig. 6g), leading to retrotransposon silencing and tumor immune evasion.

### PHF8 expression is anti-correlated with antiviral immune responses in human colorectal tumors

We next investigated whether our observations made in mouse cells could be recapitulated in human cells. *PHF8* depletion induced the expression of dsRNA or dsDNA sensing components, interferons and ISGs in human HT-29 and LoVo colorectal cells (Fig. 7a, b). We then analyzed *PHF8* expression using datasets derived from the TCGA database. We found that *PHF8* was highly expressed in tumors compared with adjacent normal tissues in colorectal adenocarcinoma patients (Fig. 7c). Moreover, *PHF8* overexpression was associated with lower overall survival (Fig. 7d).

Next, we analyzed the correlation of *PHF8* expression and human ERV expression in TCGA colorectal adenocarcinoma cohorts[39] and Mendeley Data[40]. We found that many ERVs were enriched in *PHF8*-low group (Supplementary Fig. 7a). In particular, *LTR9C* and *LTR2B* expression were anti-correlated with *PHF8* expression (Supplementary

Fig. 7b), suggesting that PHF8 repressed specific ERVs in human colorectal adenocarcinoma. We also tested these results in a series of human colorectal tumor cell lines using the CCLE database and Mendeley Data[40], and consistently found that *PHF8* expression was negatively correlated with human ERV expression (Supplementary Fig. 7c).

Given that PHF8 depletion activated human ERV expression and subsequent antiviral responses in murine and human colorectal tumor cells, we further explored the correlation of *PHF8* expression and antiviral responses using RNA-seq data derived from the TCGA database. KEGG analysis revealed that several immune-related pathways, including cytokine-cytokine receptor interaction, virus infection, and host defense pathways, were enriched in the *PHF8* low expression group (Fig. 7e). The interferon and inflammatory responses were among the leading upregulated pathways in *PHF8* low expression patients according to GSEA analysis (Fig. 7f, g). Furthermore, *PHF8* mRNA levels negatively correlated with *IFNG*, *TLR3*, *IFIH1*, *STAT1* and *OASL* expression (Fig. 7h), suggesting that PHF8 restrains adaptive immune responses in human colorectal tumors. We also analyzed *PHF8* expression levels of responders and non-responders in ICB therapy cohorts. In both the ICB clinical cohorts of melanoma treated with combined anti-PD-1 and anti-CTLA-4[41] and gastric cancer treated with anti-PD-1[42], responders showed significantly lower *PHF8* expression levels than non-responders, suggesting that *PHF8* expression was anti-correlated with ICB responses (Fig. 7i).

### Discussion

Colorectal cancer, which shows a high degree of heterogeneity, is one of the major killer diseases worldwide. As the majority of patients with colorectal cancer are not responsive to ICB therapy[43], there is an urgent need for new therapeutic approaches to enhance anti-tumor immunity and augment the antitumor immune responses. Mutations in PHF8 play critical roles in brain development and mental disease[13,16,44]. PHF8 is also aberrantly expressed in hematologic and solid tumors[19,20]; however, its role in regulating the tumor microenvironment are poorly understood. In this study, we functionally validate the relationship between tumoral PHF8 and anti-tumor immunity in multiple colorectal tumor models. PHF8 loss promotes the development of an anti-tumor immune memory, inflames immunologically-cold tumors, and sensitizes ICB-based immunotherapy.

In spite of an immune-suppressive role of PHF8 in colorectal tumor-bearing models, it is critical to dissect how PHF8 impairs tumor immunity. Epigenetic regulators have been recently implicated in immune escape and immunotherapy sensitivity[45] through a mechanism of a viral mimicry response[46–48], representing an attractive approach to directly target epigenetic factors or to combine epigenetic therapy with immunotherapy. In this study, we found that PHF8 loss in tumor cells removes an epigenetic checkpoint that restrains antiviral immune responses, endogenous nucleic acid sensing and tumor inflammation, leading to an effective immune response against cancer cells. Our study, together with others[9–11,49,50], highlights an important role of epigenetic regulators in harnessing tumor cells and

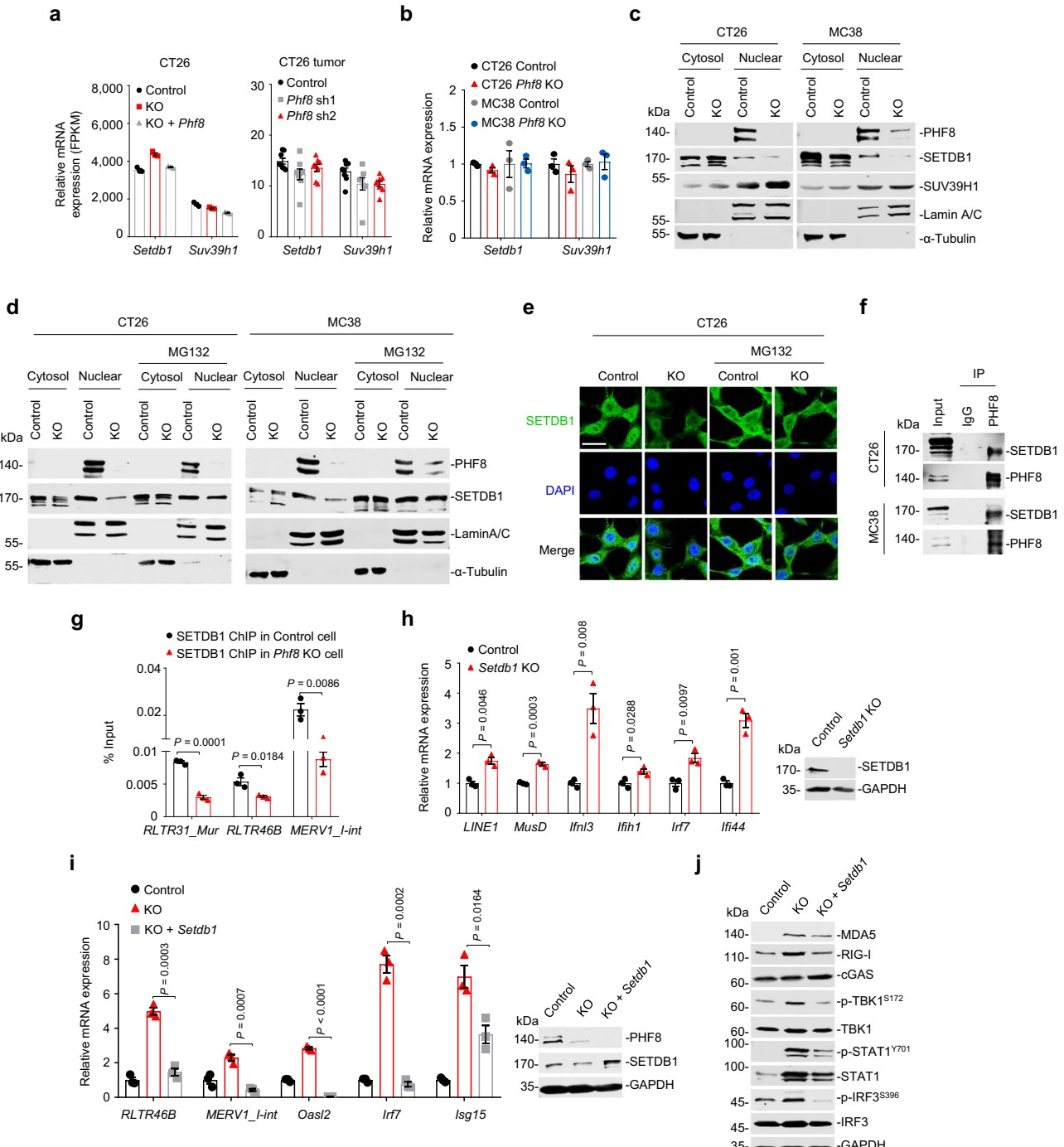

**Fig. 6 | Depletion of *PHF8* activates retrotransposons by destabilizing SETDB1.**
**a** *Setdb1* and *Suv39h1* expression (FPKMs) in the control, *Phf8* KO and *Phf8* KO + *Phf8* CT26 cells (*left, n = 3*), and the tissues of the vector control and sh*Phf8* CT26 tumors (*right, n = 8*). **b** RT-qPCR analysis of *Setdb1* and *Suv39h1* in the control and *Phf8* KO CT26 or MC38 cells. **c** Immunoblot analysis for PHF8, SETDB1 and SUV39H1 expression in the nuclear and cytoplasmic fractions. Lamin A/C or α-Tubulin was used as protein loading controls for the nuclear and cytoplasmic fractions, respectively. **d** Western blot analysis of SETDB1 and PHF8 in cytosolic and nuclear extracts from the control or *Phf8* KO cells treated with dimethyl sulfoxide (DMSO) or 15 μM MG132 for 6 hours. **e** Endogenous SETDB1 shows cytoplasm and nuclear localization. CT26 control or *Phf8* KO cells were treated with DMSO or 15 μM MG132 for 6 h, and then analyzed with immunofluorescent staining of SETDB1 (green) and DAPI (blue). Representative images of 3 independent experiments. Scale bar,

50 μm. **f** Western blot analysis of PHF8 immunoprecipitates of whole cellular extracts from CT26 and MC38 cells. **g** ChIP–qPCR showing binding patterns of SETDB1 on retrotransposons in the control and *Phf8* KO CT26 cells. **h** RT-qPCR analysis of retrotransposons and ISGs in the control and *Setdb1* KO CT26 cells. Western blot results showing *Setdb1* knockout efficiency. **i** RT-qPCR analysis of retrotransposons and ISGs in *Phf8* KO CT26 cells expressing *Setdb1* (pcDNA3.1-*Setdb1*) (*left*). Western blot analysis showing *Setdb1* overexpression in *Phf8* KO CT26 cells (*right*). **j** Western blot analysis showing DNA and RNA sensors as well as interferon response related protein expression. For **b**, **g**, **h**, and **i**, values are expressed as mean ± SEM. *n* = 3 biologically independent samples. Unpaired two-sided Student's *t*-test in **g**–**i**. The immunoblots in **c**, **d**, **f**, **h**, **i**, and **j** are representative of three independent experiments. Source data are provided as a Source Data file.

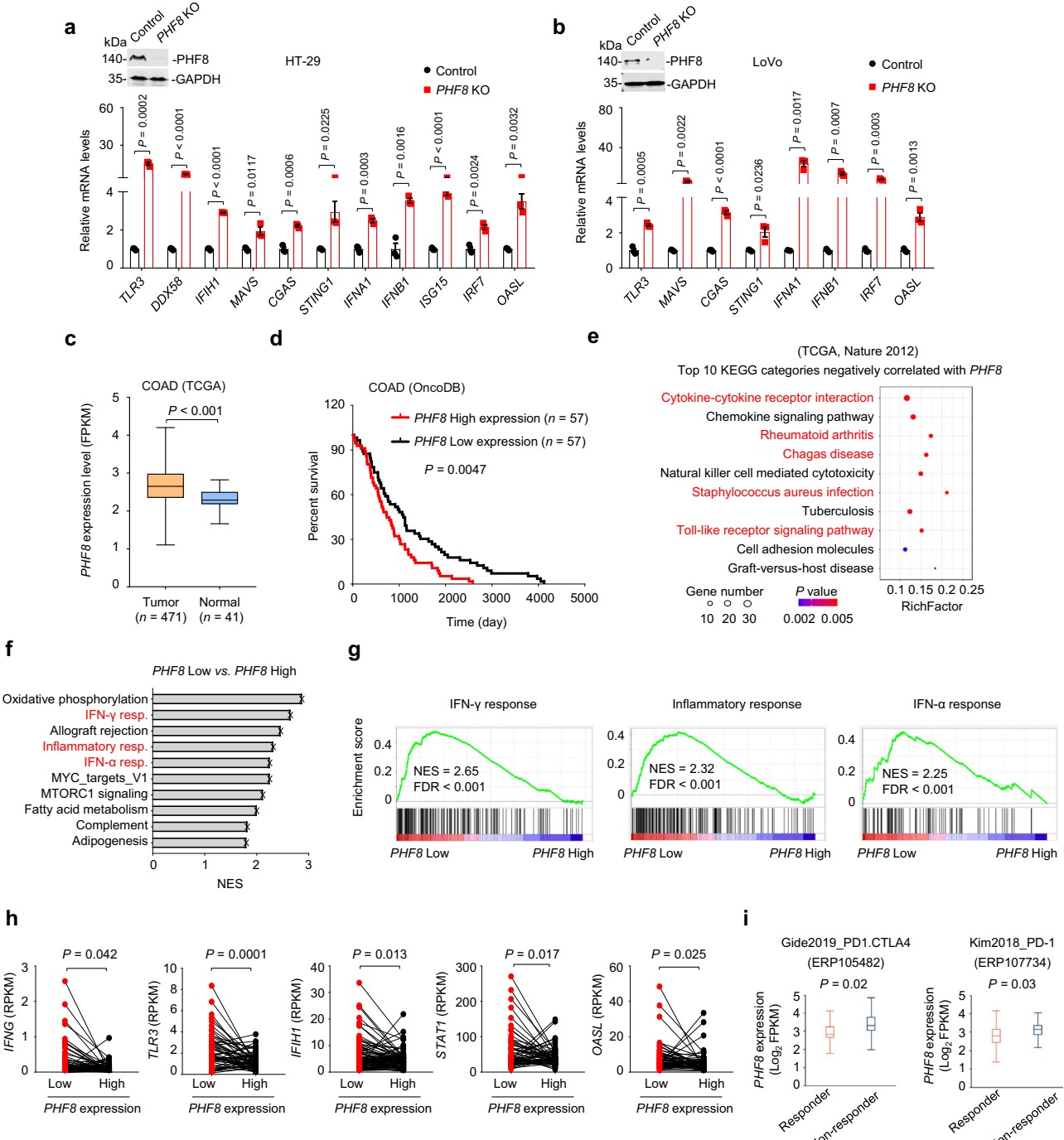

**Fig. 7 | PHF8 depletion activates antiviral immune responses in human tumors.** **a**, **b** RT-qPCR analysis of transcripts of selected IFNs and ISGs in the vector control, *PHF8* KO HT-29 **a** and LoVo **b** cells. Western blot results showing *PHF8* knockout efficiency. **c** The analysis of *PHF8* mRNA expression levels in the tumor (*n* = 471) and adjacent normal tissues (*n* = 41) from TCGA colorectal adenocarcinoma patients. Tumor, Min = 1.11, Q1 = 2.36, Med = 2.65, Q3 = 2.98, Max = 4.2, Upper Whisker = 3.91, *n* = 471; Normal, Min = 1.66, Q1 = 2.19, Med = 2.29, Q3 = 2.47, Max = 2.56, Upper Whisker = 2.91, *n* = 41. **d** Overall survival of *PHF8*-high and *PHF8*-low colorectal adenocarcinoma patients assessed by Kaplan–Meier curve analysis using OncoDB database. *n* = 57 patients per group. **e** Top 10 KEGG pathways that negatively correlated with *PHF8* expression in the TCGA datasets of colorectal adenocarcinoma. **f** Gene set enrichment analysis by comparing RNA-seq data from "*PHF8* low" group (lowest quartile) with that from "*PHF8* high" (highest quartile) group. Shown are selected top 10 upregulated pathways (FDR < 0.05). *n* = 3. NES, *nor*malized enrichment score. **g** Gene set enrichment analysis of RNA-seq data showing IFN-γ (*left*), inflammatory response (*middle*) and IFN-α (*right*) that were upregulated

in "*PHF8* low" group (lowest quartile) compared with "*PHF8* high" (highest quartile) group. *n* = 3. **h** The expression levels of *IFNG*, *TLR3*, *IFIH1*, *STAT1*, and *OASL* were higher in "*PHF8* low" group (lowest quartile) than that in "*PHF8* high" (highest quartile) group. **i** *PHF8* mRNA expression levels of responder and non-responder in ICB treatment clinical trials with combined treatment of anti-PD-1 and anti-CTLA4 in melanoma (ERP105482, *left*) or anti–PD-1 therapy in gastric cancer (ERP107734; *right*). Left: Responder, Min = 1.78, Q1 = 2.65, Med = 2.82, Q3 = 3.23, Max = 4.11, Upper Whisker = 4.1, *n* = 30; N*on*-responder, Min = 1.98, Q1 = 3.05, Med = 3.3, Q3 = 3.76, Max = 4.83, Upper Whisker = 4.83, *n* = 11. Right: Responder, Min = 1.36, Q1 = 2.42, Med = 2.81, Q3 = 3.13, Max = 4.19, Upper Whisker = 4.2, *n* = 12; N*on*-responder, Min = 2.15, Q1 = 2.87, Med = 3.17, Q3 = 3.35, Max = 4.07, Upper Whisker = 4.07, *n* = 33. For **a** and **b**, values are expressed as mean ± SEM. *n* = 3 biologically independent samples. Unpaired two-sided Student's *t*-test in **a**–**c**, **h**, and **i**, and log-rank test in **d**. The immunoblots in **a**, **b** are representative of three independent experiments. Source data are provided as a Source Data file.

their behavior in the context of an intact immune system. It is noteworthy that host defense pathways and immune-related signatures are also highly enriched in the *PHF8* low expression group of human tumors. *PHF8* overexpression is associated with lower overall survival of colorectal cancer patients. Moreover, *PHF8* expression anti-correlated with human endogenous retrovirus expression and ICB responses. Our study provides insights towards the development of small-molecule inhibitors or proteolysis-targeting chimeras that pharmacologically target PHF8, which are expected to induce viral mimicry and effective immune response.

We uncover that PHF8 enables immune evasion by silencing retrotransposons. Ablation of PHF8 function in colorectal tumor cells induces the expression of a cluster of transposable elements and releases immunostimulatory nucleic acids, ultimately leading to the activation of interferon signaling and antigen processing and presentation. As the first discovered histone H3K4 demethylase, lysine-specific demethylase 1 (LSD1, KDM1A) represses endogenous retrovirus expression reliant on H3K4me2 demethylation. Disruption of LSD1 catalytic activity stimulates a TLR3- and MDA5-dependent anti-tumor T cell immunity[10]. Our rescue results from the catalytically inactive mutant (*Phf8*[H247A]) showed that PHF8 in silencing retrotransposon and a viral mimicry response and even in tumor immune evasion was independent of its catalytic activity. Asensio-Juan and colleagues discovered that PHF8 bound to a subset of IFNγ-responsive gene promoters and kept the promoters in a silent state through its association with HDAC1 and SIN3A[51]. This effect is coupled with low levels of H4K20me1, indicating that PHF8 may function as a transcriptional repressor dependent on its catalytic activity. The plausible mechanisms of PHF8 in regulating retrotransposon expression upstream and inflammatory gene expression downstream may suggest that PHF8-loss elicits immune responses at multiple levels, which is likely determined by specific biological contexts.

PHF8 is specifically expressed in the nucleus, while SETDB1 is distributed in both the cytoplasm and nucleus. Mechanistically, PHF8 directly interacts with SETDB1 in the nucleus, independent of ATF7IP's regulation. PHF8 silences H3K9me3-marked retrotransposons by protecting the H3K9 methyltransferase SETDB1 from proteasomal degradation in the nucleus rather than direct binding to transposable elements. This mechanism of action of PHF8 is similar to that of the histone demethylase KDM5B, which also regulates SETDB1 stability in the nucleus to silence retrotransposons in a demethylase-independent manner[11]. The SETDB1-containing KAP1 complex is known to restrain transposable elements during genome evolution[52]. Accordingly, SETDB1 loss de-represses retrotransposons to encode viral proteins and trigger T cell responses[9]. Given that transposable elements are regulated by stage- and/or context-specific patterns, PHF8-associated immune modulation is likely to be pleiotropic and context-dependent. Further biochemical and cellular experiments are needed to clarify how PHF8 maintains the stability of nuclear SETDB1 and further regulates its dynamic activity in tumor cells.

Overall, our findings identify the underlying mechanism by which PHF8 decreases colorectal tumor immunogenicity and suggest that targeting PHF8 is a promising viral mimicry-inducing approach to enhance intrinsic anti-tumor immunity and further obliterate colorectal tumor growth.

## Methods

All mouse experiment procedures were approved by the Institutional Animal Care and Use Committee of East China Normal University and conducted in accordance with the guidelines (protocol number AR2021-265).

### Cell lines

CT26 and MC38 mouse colon carcinoma cells, 4T1 mouse breast carcinoma cells, 293 T cells, HT-29 and LoVo human colon carcinoma cells were purchased from the American Type Culture Collection (ATCC; Manassas, VA). KPC cells were isolated from mouse PDAC tumors driven by mutant *Kras* and mutant *Trp53* and gifted from Dr. Zhigang Zhang (Shanghai Jiao Tong University, Shanghai, China). Cells were cultured in Dulbecco's modified Eagle medium (DMEM, Sigma Aldrich, St Louis, MO) supplemented with 10% fetal bovine serum (FBS, Thermo Fisher Scientific, Waltham, MA), 100 units per milliliter of penicillin and 100 μg mL$^{-1}$ streptomycin (Thermo Fisher Scientific). All the cell lines were authenticated by the short tandem repeat method and tested negative for mycoplasma.

### CRISPR-Cas9-mediated gene knockout

Stable *Phf8* knockout (KO) and further DNA- or RNA-sensor and adaptor KO (*Cgas*, *Ddx58*, *Ifih1*, *Sting*, and *Mavs*) cell lines were generated using the lentivirus-mediated CRISPR-Cas9 technology[53]. The sgRNA oligos for target genes were annealed and cloned into the *Bsm*B1 (Thermo Fisher Scientific) digested plasmid lentiCRISPR v2 vector (52961, Addgene, Cambridge, MA). To knockout target genes, CT26 and MC38 cells were transiently transfected with lentiCRISPR-v2 vector carrying respective sgRNAs, and selected with puromycin (puro; 5 μg mL$^{-1}$ for MC38 and CT26 cells) or neobiotic (neo; 2.5 mg mL$^{-1}$ for MC38 and CT26 cells) at super-low density in 96-well plates for 7-10 days, as described previously[54]. Colonies were amplified and validated for KO by immunoblots. For double KO, *Phf8* KO CT26 or MC38 cells were used to deplete the second target gene as described above. A list of sgRNAs is provided in Supplementary Data 9.

### Gene knockdown by shRNA

For stable knockdown of *Phf8*, shRNA oligos against *Phf8* were annealed and cloned into the pLKO.1-TRC cloning vector (10878, Addgene). Lentiviruses carrying pLKO.1-TRC plasmids were produced by co-transfecting 293 T cells with two helper plasmids (pSPAX2 and pMD2G). The packaged viral supernatant was harvested by passing through 0.45 μm filter 72 h after transfection. Infected CT26 and MC38 cells were selected with puro as described above. The shRNA oligo sequences for their respective target genes are listed in Supplementary Data 9.

### Molecular cloning

The mouse *Phf8* cDNA (NM_001113354) was PCR-amplified from complementary DNA samples derived from MC38 cells. Mouse *Phf8* sequences were cloned into the lentiviral vector pLVX-CMV-EF1-neo (Addgene) for gene ectopic expression. The PCR primers used to clone mouse *Phf8* are listed as follows: *XhoI-Phf8 forward*, 5′-CTCGAGAT GGCCTCGGTGCCTGTGTATTG-3′; *BamHI-Phf8 reverse*, 5′-GGATCCGC GGCCGCTCTAGAACTAGTTCACAGAAGTAACTTG-3′. The *Phf8* H247A catalytically inactive mutant was generated using a QuikChange Lightning Site-Directed Mutagenesis Kit (Stratagene, La Jolla, CA). Stable *Phf8* re-expression in *Phf8* KO or the vector control cells was generated as described in the previous section.

### Western blot analysis

Western blotting assays were performed using a standard methods as described previously[55,56]. Whole-cell lysates were prepared using radio-immunoprecipitation assay buffer containing protease inhibitor cocktail (Sigma Aldrich) and phosphatase inhibitor cocktail (Sigma Aldrich). The protein concentration of cell lysates was assayed using a BCA Protein Assay kit (Thermo Fisher Scientific). Samples in SDS loading buffer were heated for 10 min at 100 °C and loaded onto 10% or 15% SDS−PAGE gels. Membranes were blocked with 5% bovine serum albumin for 1 h at room temperature and incubated with respective primary antibodies overnight at 4 °C, followed by secondary antibody incubation for 1 h at room temperature. The membranes were visualized using a fluorescent western blot imaging system (Odyssey Biosciences, LI-COR, https://www.licor.com/). The primary antibodies used

are listed as follows: anti-PHF8 (ab280887, 1:1000) antibody was purchased from Abcam (Cambridge, MA). Anti-RIG-I (#3743, 1:1000), anti-MDA5 (#5321, 1:1000), anti-MAVS (#4983, 1:1000), anti-cGAS (#31659, 1:1000), anti-p-TBK1 (#5483, 1:1000), anti-TBK1 (#3504, 1:1000), anti-p-IRF3 (#4947, 1:1000), anti-IRF3 (#4302, 1:1000), anti-SUV39H1 (#8729, 1:1000), anti-α-Tubulin (#2125, 1:1000), anti-Lamin A/C (#4777, 1:1,000), anti-p-STAT1 (#9167, 1:1000) and anti-STAT1 (#9172, 1:1000) antibodies were purchased from Cell Signaling Technology (CST, Danvers, MA). Anti-GAPDH (AB0036, 1:1000) antibody was obtained from Abways (Shanghai, China). Anti-SETDB1 (11231-1-AP, 1:1000) antibody was obtained from Proteintech (Chicago, IL). Anti-ATF7IP (sc-166753, 1:1,000) was obtained from Santa Cruz Biotechnology (Santa Cruz, CA).

## RNA extraction and RT-qPCR
Total cellular RNA was extracted using TRIzol extraction (Invitrogen, Eugene, OR) according to the manufacturer's instructions. One μg of total RNA was reverse transcribed into cDNA using a PrimeScript RT Reagent kit (Takara Biotechnology, Dalian, China). The obtained cDNA samples were diluted and subjected to real-time quantitative PCR (RT-qPCR) assays using SYBR Premix Ex Taq (Takara Biotechnology). The primer sequences for target genes are listed in Supplementary Data 10. The 2 − ΔΔCT method was used to calculate relative gene expression. RT-qPCR data were normalized to GAPDH and presented as fold change (tested samples over the control).

## Cell viability assays
The vector control and *Phf8* KO cells were seeded onto 96-well plates at a density of 2000 cells per well and allowed to adhere overnight. Cell viability was measured using the sulforhodamine B colorimetric assay at indicated time points. The average $OD_{515}$ of the vector control cells was set to 100%, and the percentage of viable cells was calculated accordingly.

## Two-dimensional clonogenic assays
The vector control and *Phf8* KO cells were plated onto 6-well plates at a density of 2000 cells per well and allowed to adhere overnight. The cells were then cultured in complete media for 7–10 days. The medium was replaced every other day. Cells were fixed with 4% formaldehyde, stained with 0.5% crystal violet and photographed.

## Animal experiments
Mice were housed at an ambient temperature of 72 °F, with a humidity of 30–70%, and a light cycle of 12 h on/12 h off set from 7 am to 7 pm in the East China Normal University Animal Facility and randomly divided into groups. To avoid potential impacts of physiological cycle of female mice on experimental results, we chose male mice for our animal study. Six-8 week-old male C57BL/6 mice, male Balb/c mice, and male nude mice were purchased from the National Rodent Laboratory Animal Resources (Shanghai, China). Six-8 week-old RAG2$^{-/-}$ male mice (C57BL/6 background) were gifted by Dr. Bing Du (East China Normal University, Shanghai, China). Prior to experiments, mice were allowed one week adaptation to housing conditions. CT26 and MC38 cells were injected under the dorsal skin of mice at doses of $5 \times 10^5$ and $1 \times 10^6$ per inoculate, respectively, in a total volume of 100 μL. Tumor volume (0.5 × length × width$^2$) was measured with calipers at indicated time points. After tumor collection, tumors were weighed and samples were collected for flow cytometric analysis, immunofluorescence staining, and immunohistochemistry. For tumor rechallenge experiments, wild-type C57BL/6 or Balb/c mice were first inoculated with $5 \times 10^5$ CT26, $1 \times 10^6$ MC38 *Phf8* KO cells, or PBS alone. After 38 days, the same mice were inoculated with $1 \times 10^6$ CT26, $2 \times 10^6$ MC38 or $5 \times 10^5$ 4T1 cells, respectively. The tumor volume was measured as described above. For immunotherapy treatments,

mice were treated with 50 μg (CT26) anti-PD-1 antibodies (clone RMP1-14, 114101, BioLegend, San Diego, California) or IgG2b isotype control twice per week for a total of 2 weeks after tumor injection for CT26 cells on day 6. Our animal protocol sets the maximal tumor volume at 2000 mm$^3$. Percent survival was determined by tumor volume larger than 2000 mm$^3$ or a humane endpoint. Mice were humanely euthanized by carbon dioxide inhalation, followed by cervical dislocation.

## Flow cytometry
Approximately $5 \times 10^5$ the vector control and sh*Phf8* CT26 cells were subcutaneously inoculated into Balb/c mice. The tumor tissues were removed, minced and then incubated in DNase collagenase I (SCR103, Sigma Aldrich) and collagenase IV (C4-BIOC, Sigma Aldrich) at 37 °C for 20–30 min. Tumor cells were then passed through a 70-μm cell strainer to obtain a single-cell suspension. Cell suspension was stained with Zombie AquaTM fixable viability dye (423101, BioLegend) and surface antibodies for 30 min on ice and fixed with 1% PFA before data acquisition. The following commercial antibodies were used: anti-CD45-PerCP/Cyanine5.5 (clone 30-F11, 103131, BioLegend), anti-CD8a-APC (clone 53-6.7, 100711, BioLegend), anti-CD62L-PE (clone MEL-14, 144407, BioLegend), and anti-CD44-PE/Cyanine7 (clone IM7, 103030, BioLegend).

## In vivo competition assays
CT26 or MC38 cells stably expressing green fluorescent protein (GFP) or mCherry were infected with *Phf8*-targeting sgRNA or control lentiviral vectors, respectively, and selected with 5 μg mL$^{-1}$ puro for 7–10 days. Subsequently, approximately $2.5 \times 10^5$ CT26 or $5 \times 10^5$ MC38 *Phf8* KO cells (GFP-expressing) and equal amount of their control cells (mCherry-expressing) were mixed and inoculated subcutaneously into Balb/c or C57BL/6 male mice. Tumors were excised 12 days after inoculation and digested with DNase I, collagenase I and collagenase IV at 37 °C for 20–30 min. The dissociated tumor cells were filtered, washed, and resuspended in ice-cold PBS with 1% FBS. The ratio of GFP and mCherry tumor cells was determined using BD FACSDiva (V.8.0.1), and quantitative analysis was carried out using FlowJo (v10) (https://www.flowjo.com/solutions/flowjo).

## RNA-seq and TE RNA-seq analysis
Purified total RNA from cells or tumor tissues were isolated using the TRIzol reagent (Invitrogen). Poly (A)-enriched sequencing libraries were constructed using the NEBNext® Ultra™ II Directional RNA Library Prep kits (New England Biolabs, Ipswich, MA). Library sequencing was performed by Novogene (Tianjin, China) using an Illumina HiSeq 4000 platform.

RNA-seq data processing was performed as described previously[9]. To analyze gene expression, reads were aligned to the reference transcriptome using RSEM[57] and bowtie2 (v2.3.4)[58] (-bowtie2-bowtie2-sensitivity-level very_sensitive-no-bam-output-estimate-rspd), and the index was built by RSEM with the mouse genome, mm10, and Enemble gene annotation track v.74. For TE analysis, the reads were mapped to the mouse genome (mm10) using the STAR aligner (v2.5.4b)[59] and the counts for each gene or TE family were counted using scTE[9]. DESeq2 (v1.20.0)[60] was used for data normalization and differential expression analysis. Differentially expressed genes and TEs were defined by a Benjamini-Hochberg-corrected *P* value < 0.05 and an absolute fold-change > 2 (for genes).

## ChIP analysis
The chromatin immunoprecipitation (ChIP) assays were performed using the Simple ChIP Plus Enzymatic Chromatin Immunoprecipitation kits (agarose beads) (#9004, CST) according to the manufacturer's instructions. Anti-H3K9me1 (ab9045, 1:1,000), anti-H3K9me2 (ab176882, 1:1,000), anti-H3K27me2 (ab24684, 1:1,000),

and anti-H4K20me1 (ab177188, 1:1,000) antibodies were obtained from Abcam. Anti-H3K9me3 (#13969, 1:1,000) antibody was purchased from CST. ChIP–qPCR analysis was performed using primers described in Supplementary Data 10. For ChIP–seq experiments, 4 μg spike-in chromatin and 2 μg spike-in antibody were added into each ChIP reaction. ChIP–seq libraries were prepared using the NEBNext Ultra II DNA Library Prep kits for Illumina (E7103, NEB), and sequenced using Illumina NovaSeq S4 2×150 paired-end sequencing.

## ATAC-seq library preparation and sequencing

The vector control or *Phf8* KO CT26 cells were treated with or without 20 ng/mL IFN-γ for 24 hours and then dissociated. A total number of 50,000 cells were washed twice in cold PBS at 4 °C and resuspended in 50 μL cell lysis buffer (10 mM Tris-HCl pH 7.4, 10 mM NaCl, 3 mM $MgCl_2$, 0.05% NP40) on ice for 5 min. Samples were washed with 950 μL cold wash buffer (10 mM Tris-HCl pH 7.4, 10 mM NaCl, 3 mM $MgCl_2$, 0.05% NP40, 0.1% Tween 20) and subsequently centrifuged at 500 g at 4 °C for 5 min. After that, supernatant were removed. The transposase reaction was carried out in 50 μL TD Buffer (10 mM Tris-HCl, pH 8, 5 mM Magnesium Chloride) supplemented with 4 μL transposase (N248, Illumina Nextera Kit, Novoprotein, shanghai, China), and incubated at 37 °C for 30 min, followed by the addition of 10 μL of 100 mM EDTA. Samples and DNA were recovered using Tagment DNA extract beads (N245, Illumina Nextera Kit, Novoprotein). Libraries were generated by PCR in 50 μL reaction system (35 μL sample, 10 μL 5x AmpliMix, 2.5 μL of each custom Illumina primers at 10 μM). The PCR program was set as follows: 72 °C for 3 min, 98 °C for 30 s, followed by 9 cycles of 98 °C for 15 s, 60 °C for 15 s, 72 °C for 8 s, 72 °C for 2 min. DNA was then recovered using DNA clean beads (N240, Illumina Nextera Kit, Novoprotein). The libraries were sequenced to 30 million reads per sample on NovaSeq 6000 S4 platform using Nexrtera Sequencing Primers.

## ChIP-seq and ATAC-seq data analysis

ChIP-seq and ATAC-seq reads were aligned to the mouse mm10 genome using bowtie2 (v2.3.4)[58] with the options '-p 20 –very-sensitive –end-to-end –no-unal–no-mixed -X 2000'. To analyze repetitive sequences, only the best alignment was reported for multi-mapped reads; if more than one equivalent best alignment was found, one random alignment was then reported. Reads mapping to unassigned sequences were discarded. For unique alignments, duplicate reads and low-mapping-quality reads were filtered using Picard (https://broadinstitute.github.io/picard/) and SAMtools (v.1.9). ChIP-seq enriched peaks were called by MACS2 (v.2.1.2)[61] and SICER2[62], and the common peaks were merged using bedtools (v2.27.1) and used for subsequent analysis. ATAC-seq peaks were called using DFilter (with the settings: -bs = 100 –ks = 60 –refine)[63]. The enriched peaks were annotated using the annotatePeaks.pl script from the Homer package[64]. For visualization, BAM files of biological replicates were merged using SAMtools (v.1.9)[65]. BigWig files were generated using deeptools (v3.0.2)[66] using the RPKM normalization method. Figures illustrating these continuous tag counts over selected genomic intervals were created using the IGV browser[67]. The signals for TEs were processed as described previously[41]. Coordinates and annotations of TEs were downloaded from UCSC Genome Browser (mm10) version of RepeatMasker (http://www.repeatmasker.org). For TE enrichment analysis, we divided the TEs into evenly spaced 500 bp bins. The coverage signal on each bin was extracted using deeptools[47]. The bin with the maximum signal was used as the observed immunoprecipitation value, which was then divided by normalized read counts of a matched input sample and expressed as $\log_2$ (fold enrichment). For the randomly expected background value, the coverage signal of random genomic regions of the same size and number of the TE type was measured. TEs with observed IP values smaller than the randomly expected background value were excluded from analysis. Other analysis was performed using glbase3[68].

## Patient dataset analysis

For colorectal adenocarcinoma TCGA cohorts, patients were divided into *PHF8* low and high subgroups according to the mRNA expression level of *PHF8*. Specifically, the 25% quantile and 75% quantile of the normalized *PHF8* expression were defined as the cutoffs. Tumor samples with normalized *PHF8* gene expression lower than or equal to the 25% quantile were classified into the *PHF8* low group while higher than or equal to the 75% quantile were classified into the *PHF8* high group. Differential gene expression analysis were performed by comparing the *PHF8* high group to the *PHF8* low group. The colorectal adenocarcinoma TCGA cohorts were profiled by RNA-seq. The differential gene expression analysis identified 877 unique upregulated genes in the *PHF8* high subgroup ($\log_2$ FoldChange > 1 and adjusted P value < 0.05) for the colorectal adenocarcinoma. WebGestalt 2019[69] was used to identify the GO pathways of the 877 genes negatively correlated with *PHF8* mRNA expression from the TCGA colorectal adenocarcinoma cohorts.

We studied the clinical relevance of PHF8 regulating ERV expression by analyzing the association of *PHF8* expression with overall survival or with immunotherapy responses in clinical trials. For survival analysis, the Kaplan-Meier survival estimate was used to test the association between *PHF8* expression and patient survival in colorectal adenocarcinoma patients from OncoDB[70]. Colorectal adenocarcinoma patients were divided into *PHF8*-low and *PHF8*-high subgroups according to the normalized *PHF8* mRNA expression levels at the cutoffs of the 25% quantile and 75% quantile. In immunotherapy trials, we collected and analyzed two cancer patient cohorts with available data of RNA-seq and immunotherapy responses from published studies[41,42] and ICBatlas[71]. ERV expression dataset normalized by variance stabilizing transformation (VST) was downloaded from Ito et al. on Mendeley Data[40] and used for correlation analysis. Scaled VST values were plotted for expression heatmap.

## Immunohistochemistry and immunofluorescence

Immunohistochemistry and immunofluorescence were performed as described previously[72,73]. For immunohistochemistry analysis, tumor tissues from mice were fixed in 10% buffered formalin (Thermo Fisher Scientific) at 4 °C overnight, paraffin-embedded and sectioned and then mounted. Sections were deparaffinized in xylene, rehydrated and washed in PBS. Subsequently, sections were boiled with antigen unmasking solution (H-3300, Vector labs) for 20 min, blocked with 10% normal goat serum in PBS at room temperature for 1 h, and stained by standard procedures using antibodies against mouse PHF8 (GB114477, Servicebio, Wuhan, China).

For immunofluorescence was performed as described previously[74]. For immunofluorescence, 4 μm paraffin sections of tumor tissues were baked at 60 °C for 2 h, and then deparaffinized. Antigen was retrieved at EDTA antigen retrieval buffer (pH 8.0) and maintained at a sub-boiling temperature for 8 min, standing for 8 min and then another sub-boiling temperature for 7 min. For cell staining in culture, cells were fixed in 4% paraformaldehyde for 15 min, washed with PBS buffer for 3 times, and then blocked with 3% BSA. Cells were stained by standard procedures using antibodies. Images were captured by use of confocal microscopy. The following antibodies were used: anti-dsRNA monoclonal antibody J2 (10010200, SCICONS), anti-dsDNA (MAB1293, Merck), anti-SETDB1 (11231-1-AP, Proteintech), anti-CD8a (GB13429, Servicebio), anti-IFN-γ (GB11107-1, Servicebio) and anti-GZMB (AF0175, Affinity, Shanghai, China) antibodies.

## Statistical analysis

Statistical analysis was performed using GraphPad Prism (v8) software (http://www.graphpad.com/) and Microsoft Excel (v15). P-values of less

than 0.05 were considered statistically significant. Two-way ANOVA was used for multiple comparisons in tumor growth experiments. Log-rank tests were used for mouse survival analysis. In other experiments, comparisons between two groups were made with unpaired two-sided Student's *t*-tests. No statistical methods were used to predetermine sample size. Western blot analysis, immunofluorescence staining, and qPCR analysis were repeated at least twice with consistent results. Animal experiments were repeated as indicated in the figure legends. Data on *PHF8* gene expression and patient survival were obtained from cBioportal (http://www.cbioportal.org/)[75] and OncoDB (https://oncodb.org/)[70]. Datasets of *PHF8* gene expression and immunotherapy responses were obtained from published studies[41,42] and ICBatlas (http://bioinfo.life.hust.edu.cn/ICBatlas)[71]. Data of ERV expression were obtained from Ito et al. on Mendeley Data (https://data.mendeley.com/datasets/c7r7dw9p42/1)[40].

### Reporting summary
Further information on research design is available in the Nature Portfolio Reporting Summary linked to this article.

## Data availability
All genomic sequencing data that support the findings generated in this study have been deposited in the Gene Expression Omnibus database under the accession GSE212779 and GSE211526. The publicly available data on gene expression of colorectal adenocarcinoma were obtained from cBioportal[75] (http://www.cbioportal.org). The publicly available data on overall survival of colorectal adenocarcinoma were obtained from OncoDB[70] (https://oncodb.org). The publicly available data on *PHF8* gene expression and immunotherapy response were obtained from ICBatlas[71] (http://bioinfo.life.hust.edu.cn/ICBatlas). The publicly available data on ERV expression were obtained from Mendeley Data[40] (https://data.mendeley.com/datasets/c7r7dw9p42/1). The publicly available data of H3K9me3 ChIP-seq in the control and SETDB1 KO B16 melanoma cells were downloaded and reanalyzed from the Gene Expression Omnibus database under the accession GSE155972[9]. The remaining data are available with the Article, Supplementary Information or Source data file. Source data are provided with this paper.

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

## Acknowledgements

This work was sponsored by the National Natural Science Foundation of China (project number 82122045 and 82073073, received by X.P.), the Natural Science Foundation of Shanghai (project number 23ZR1418900, received by X.P.), Innovative Research Team of High-level Local Universities in Shanghai (project number SHSMU-ZDCX20210802, received by X.P.), the National Key R&D Program of China (project number 2021YFA1102200, received by J.H.), the Guangdong Natural Science Funds for Distinguished Young Scholar (project number 2023B1515020111, received by J.H.), and the China Postdoctoral Science Foundation (project number 2023M731109, received by J.F.).

## Author contributions

X.P. and Y.L. initiated, conceived, and designed the study. Y.L., L.H. and Z.W. performed the main biological experiments. J.H. performed the

bioinformatics analysis with help from G.H. Y.K. assisted in experiments including with cell culture, plasmid construction, immunohistochemistry, and immunofluorescence staining assays. Y.L., L.H., Z.W. and K.Y. performed animal experiments, RNA-seq, ChIP-seq and ATAC-seq with help from Z.L., J.F. and N.L. Y.L., L.H. and Z.W. contributed equally to the work. X.P. and Y.L. wrote the manuscript. J.H., D.L., J.W., J.C. and M.L. helped to improve the manuscript. All authors read and approved the final manuscript.

## Competing interests

The authors declare no conflict of interests.
