## [Peer Review File · Nature Communications]

Loss of PHF8 induces a viral mimicry response by activating endogenous retrotransposonsReviewers' Comments:

Reviewer #1:

Remarks to the Author:

The manuscript entitled 'Loss of PHF8 induces a viral mimicry response by activating endogenous retrotransposons' by Liu et al. showed a critical role of the histone demethylase PHF8 in mediating tumor immune escape. The authors used murine colorectal models to show that PHF8 ablation abrogated tumor growth, activated anti-tumor immune memory and enhanced sensitivity to ICB therapy. They further showed that PHF8 depletion triggered a viral mimicry response dependent on endogenous nucleic acid sensing pathways. Mechanistically, they presented convincing evidence that PHF8 represses H3K9me3-dependent retrotransposons by promoting proteasomal degradation of the H3K9 methyltransferase SETDB1 in a demethylase-independent manner.

Overall, the work is very interesting and novel, and the study is thorough. This work adds new evidence to the recent advancement revealing a direct role of epigenetic machinery in silencing retrotransposons and the potential to target this mechanism to induce anti-tumor immunity. I just have a couple of suggestions for the authors to further improve their manuscript:

Major comments:

1. The authors demonstrated that anti-tumor efficacy of PHF8 deficiency was dependent on an intact immune system in mouse models of colorectal adenocarcinoma (CT26 and MC38). This is an interesting finding. However, to precisely conclude that this is a general and crucial phenomenon, the authors are suggested to address anti-tumor efficacy of PHF8 deficiency in other murine cancer models.
2. The authors propose that PHF8 loss sensitizes ICB-based immunotherapy. Since 4T1 breast cancer mouse model is reported to evade PD1 blockade (according to Sci Transl Med 2022 Jun 8;14(648):eabh1261), it would be more convincing to perform the PD-1 blockade experiments using 4T1 Phf8 knockdown tumor model.
3. It would be also important to directly show the increase in dsRNA formation using the J2 antibody in Phf8 knockdown tumor tissues.
4. The authors quantified immunofluorescence density of GZMB+ and IFN- γ + CD8+ T cells to indicate the increased anti-tumor CD8+T cell immune responses in shPhf8 CT26 tumors. The authors are also suggested to perform flow cytometry analysis to confirm this finding.

Minor comments:

1. The authors should analyze the differential expression of antiviral defense-related pathway genes using the RNA-seq data derived from CT26 cells.
2. The authors are suggested to add more discussion on the potential mechanism of SETDB1 degradation in the Discussion section.

Reviewer #2:

Remarks to the Author:

In the current study, Liu et al. investigated the role of Phf8 in tumor immunity. It is interesting that Phf8 exerts its function through destabilizing Setdb1, and the authors provide many data in mouse model and cell lines, however, some of the results are not convincing and the mechanistic insight how Phf8 regulates Setdb1 stability is not fully explored. I feel the manuscript should not be published before the concerns are addressed. My detailed comments are as below.

Major concerns:

1. In Figure 5E, the majority of H3K9me3 seemed to be repressed after Phf8 KO. The authors claimed Phf8 destabilizes Setdb1. Could the author compare H3K9me3 distribution after Phf8 KO and Setdb1

- KO? Whether the two proteins control the same regions on chromatin?
2. Does Phf8 also bind to the transposons regulated by Setdb1? Does Phf8 KO affect Setdb1 recruitment to chromatin?
 3. Fig. 6D showed that after phf8 KO, Setdb1 protein level decreased specific in nuclear in a proteasome-dependent manner. The authors further showed that Phf8 demethylase activity is not required. Could the authors show more mechanisms how Phf8 regulates Setdb1? Does Phf8 interact with Setdb1? Why does the nuclear fraction specific go down?
 4. Fig. 6I, Setdb1 expression greatly repressed the total amount of STING. Does it mean Setdb1 repress STING expression? At transcription level or at post-translation level?
 5. Recent publications showed that SETDB1 KO caused cytoplasmic DNA and histone H3. Does it also happen in Phf8-KO cells?
 6. In Figure 1M, the effects of two shRNA are not consistent. In IF staining, sh1 caused elevation of IFN- γ , while sh2 did not. In the statistic result, sh1 showed no effect on IFN- γ , which was totally different from IF. Moreover, the staining of Phf8 should be provided here.
 7. The changes of several selected REs are not consistent in Figure 4D & 4E.
 8. Based on their TCGA analysis, does PHF8 differently express in tumor tissues? Does its expression correlate with patient survival rates?
 9. In Sup Fig.1B-D, two sgRNAs were used for Phf8 KO in each cell line, why only one lane for Phf8 KO was shown in Sup Fig. 1A? For other results, why only one KO cell line was selected?

Minor points:

10. Fig. 6E, based on the intensity of DAPI staining, the pictures of Phf8 KO seemed to be less exposed.
11. In FigS1B, one of the cell lines seems not proper labelled.

Reviewer #3:

Remarks to the Author:

In this manuscript, Liu, Hu et al explore the role of PHF8 in cancer immunotherapy response, focusing on colorectal cancer models. First they demonstrate that PHF8 loss in the well-established MC38 and CT26 syngeneic mouse models does not affect cancer cell line proliferation or tumor growth in immunocompromised mice, but promotes tumor rejection in immune competent animals. Mechanistically, they identify upregulation of multiple interferon signaling components following PHF8 KO, and activation of pTBK1, pIRF3, and pSTAT1, which are suppressed by PHF8 re-introduction. Functionally, they show that cGAS and STING KO, as well as KO of MAVS and other antiviral signaling components, can suppress immunogenicity following PHF8 KO, and find that multiple ERVs and other H3K9me3 marked retrotransposons are de-repressed. They link this finding to destabilization of SETDB1 and show that SETDB1 re-introduction can repress this program. Finally they confirm that PHF8 KO upregulates interferon signaling genes in human CRC cell lines and that low PHF8 levels in tumors correlates with increased expression of these genes as well.

Overall this is a very nice set of experiments. I have several critiques though, which should be addressed.

1. In Figure 2 the pTBK1, pIRF3, and pSTAT1 blots are convincing, but I am not convinced by pSTING – this is a transient species that precedes STING degradation, and if STING were phosphorylated one should see a double band with the total STING antibody. I would remove this piece of data unless the authors can convincingly show the double band with total STING as well. Same with Fig 6I – pTBK1 as a downstream functional readout would be more convincing than pSTING.
2. For Figure 3A and C – where is the PHF8 blot proving effective KO of PHF8 with each of the other targets? For 3E and F also have they proven effective suppression of all 3 targets? Also, since they were effective individually in vitro, why didn't they just test the single KO of STING or MAVS as

compared to the double KO in vivo?

3. Regarding mechanism, prior work from Asensio-Juan et al. (PMID 28100697) has implicated PHF8 in restraining IFN γ gene expression directly involving H4K20methylation. They authors should reconcile their findings with this additional plausible mechanism.

4. Finally, simply examining levels of interferon stimulated genes in the 2 human cancer cell lines is not enough. Does PHF8 KO increase pTBK1 and pSTAT1 in these lines? Also certain human CRC lines have silenced STING (see Xia et al. PMID 26748708). Is the impact of PHF8 KO lost in those human cell lines, or can it de-repress STING, which would potentially be interesting in terms of being able to restore immunogenicity.

Reviewer #4:

Remarks to the Author:

Liu et al. identified PHF8 as a novel silencer of retroelements and showed that the downregulation of PHF8 can activate tumor immunity. Although the story that the dysregulation of endogenous retroelements can activate tumor immunity via dsRNA formation is in line with a series of recent previous studies, the manuscript is well organized and the results of intensive experiments are convincing. Furthermore, the results of the analyses using publicly available transcriptome data of human tumors support the hypothesis that the authors proposed. I have some suggestions to improve the manuscript as below.

Major concerns

1. Page 5, line 13. Please clarify the motivation that the authors decided to investigate the functions of PHF8 in anti-tumor immunity in the Introduction section.

2. Page 7, line 6. The authors did not present the analysis to assess the synergistic effect between immune checkpoint blockade (ICB) therapy and PHF8 knockdown. The authors should add a statistical analysis specifically to assess the synergistic effect or rephrase the sentence.

3. Page 8, line 1. Please add (supplemental) Figures to clarify the correspondence of Figs. 2A and 2C. More specifically, I would like to check whether the up- or downregulated GO terms in CT26 cells are also up- or downregulated in CT26 tumors, respectively (and vice versa).

4. Page 9, line 8. Please clarify why the authors specifically picked up RLTR46B, MER68B, LTR67B, RLTR13D3, MERV1_I-int, and L1Md_A. Please explain the criteria.

5. Figs. 4A and 4B. The authors should add additional analyses to assess whether retroelements upregulated by PHF8 KO are generally downregulated by the PHF8 supplement.

6. Page 9, line 25. The authors would examine the amount of dsRNA under the RTi treatments.

7. Fig.7. To further validate the proposed model using the TCGA dataset, the authors would examine the association between the (total amount of) ERV expression level and the PHF8 expression level in the TCGA (and CCLE) datasets.

The expression level of ERVs and genes in the TCGA and CCLE datasets are publicly available here: <https://data.mendeley.com/datasets/c7r7dw9p42/1>

8. Related to Fig. 7. It would be interesting to assess whether the efficacy of ICB therapy is associated with the expression level of PHF8 in publicly available clinical data.

Minor concerns

1. Page 6, line 20. Please insert a brief explanation why the authors decided to investigate immune memory here.
2. Page 6, line 27. In order to argue "PHF8 loss promotes the development of an anti-tumor immune memory", I think that the authors should compare the effect between tumors with and without PHF8. However, such comparison is difficult since the transplantation of the same amount of WT tumor cells is lethal. Therefore, the authors would change the message here.
3. Fig. 4H. I could not understand the result. Maybe the labels KO and KO + Phf8 swapped?
4. Page 9, line 22. Please add a brief description for the J2 antibody.
5. Page 14, line 2 and others. Please rephrase "endogenous retroposons" as retroposons.
6. Supplemental Table. Please provide the tables as an excel file instead of a PDF file.

We sincerely thank you and the reviewers for those valuable comments and suggestions on our manuscript “*Loss of PHF8 induces a viral mimicry response by activating endogenous retrotransposons* (reference NCOMMS-22-51008A)”. We appreciate the opportunity to resubmit our manuscript to your high-impact journal. We have added a large amount of new data and significantly improved the quality of our manuscript by revising most of the changes suggested. To facilitate your review on our revised manuscript, we have marked all changes in red in our revised manuscript. Below are the point-by-point responses to your and all reviewers’ comments.

Comments of Reviewer #1

Overall, the work is very interesting and novel, and the study is thorough. This work adds new evidence to the recent advancement revealing a direct role of epigenetic machinery in silencing retrotransposons and the potential to target this mechanism to induce anti-tumor immunity. I just have a couple of suggestions for the authors to further improve their manuscript:

Question 1: The authors demonstrated that anti-tumor efficacy of PHF8 deficiency was dependent on an intact immune system in mouse models of colorectal adenocarcinoma (CT26 and MC38). This is an interesting finding. However, to precisely conclude that this is a general and crucial phenomenon, the authors are suggested to address anti-tumor efficacy of PHF8 deficiency in other murine cancer models.

Response: We sincerely thank the reviewer for this insightful suggestion.

To further strengthen the critical role of PHF8 loss in anti-tumor immunity, we additionally set up xenograft mouse models of breast and pancreatic cancers using *Phf8*-deficient murine breast cancer cells (4T1) and pancreatic cancer cells (KPC) (please see our new data below, **A**). Our results consistently showed that knockout of *Phf8* did not impair the growth of breast and pancreatic tumor xenografts in immunodeficient hosts (**B-E**) but abrogated tumor growth in immunocompetent hosts (**F-I**). Together with our existing data from *Phf8*-KO mouse models of colorectal cancer (using CT26 and MC38 cells) in our original manuscript, we confidently believe that PHF8 loss induces a potent anti-tumor immunity.

New data

We have supplemented these new data in **Figure S1D-S1G** and **Figure S1J-S1M** of our revised manuscript. Changes in corresponding result descriptions and figure legends are marked in red.

Question 2: The authors propose that PHF8 loss sensitizes ICB-based immunotherapy. Since 4T1 breast cancer mouse model is reported to evade PD1 blockade (according to Sci Transl Med 2022 Jun 8; 14 (648): eabh1261), it would be more convincing to perform the PD-1 blockade experiments using 4T1 Phf8 knockdown tumor model.

Response: We appreciate the reviewer's insightful comments.

We agree with the reviewer's suggestion. We additionally performed sensitizing experiments using the ICB-resistant 4T1 murine breast cancer model. We found that co-treatment with anti-PD-1 and sh*Phf8* could further decrease 4T1 tumor volume (please see our new data below, **left**) and prolong mouse survival (**right**) as compared with sh*Phf8* or anti-PD1 treatment alone. Together with our existing data that *Phf8* knockdown enabled anti-PD-1 to eradicate CT26 tumors (**Figure 1N and 1O**), we clearly showed that ablation the function of PHF8 augmented sensitivity to ICB therapy.

New data

We have supplemented these new data in **Figure S1Y** and **Figure S1Z** of our revised manuscript. Changes in corresponding result descriptions and figure legends are marked in red.

Question 3: It would be also important to directly show the increase in dsRNA formation using the J2 antibody in *Phf8* knockdown tumor tissues.

Response: Thanks for this comment.

In our original manuscript, we showed that PHF8 ablation induced the formation of cytoplasmic dsRNA foci in CT26 and MC38 cells. As suggested by the reviewer, we additionally performed immunofluorescence staining for dsRNA in CT26 xenograft tumors that were isolated 18 days after tumor cell injection. Our results consistently demonstrated that dsRNA formation was increased in *Phf8*-deficient tumors compared with the control tumors (please see our new data below).

We have supplemented this new data in **Figure S4D** of our revised manuscript. Changes in corresponding result descriptions and figure legends are marked in red.

Question 4: The authors quantified immunofluorescence density of GZMB⁺ and IFN- γ ⁺ CD8⁺ T cells to indicate the increased anti-tumor CD8⁺ T cell immune responses in sh*Phf8* CT26 tumors. The authors are also suggested to perform flow cytometry analysis to confirm this finding.

Response: We appreciate this reviewer's valuable comments.

Following to the reviewer's suggestion, we additionally conducted flow cytometry analysis of T cell immunity in sh*Phf8* CT26 tumors. As expected, we consistently found a markedly higher number of tumor-infiltrating immune cells in *Phf8*-deficient CT26 tumors, including CD8⁺ T cells, activated (CD44^{hi} CD62L^{lo}) CD8⁺ T cells (new data as shown below). Along with our immunofluorescence results (**Figure 1M**), we believe that depletion of tumor PHF8 promoted adaptive anti-tumor CD8⁺ T cell immune responses, leading to significant tumor growth repression.

New data

We have supplemented the new data in **Figure S1X** of our revised manuscript. Changes in corresponding result descriptions and figure legends are marked in red.

Question 5: The authors should analyze the differential expression of antiviral defense-related pathway genes using the RNA-seq data derived from CT26 cells.

Response: We appreciate the reviewer's suggestion.

As suggested by the reviewer, we further analyzed the differential expression of antiviral defense-related genes using the RNA-seq data derived from CT26 cells. Our analysis showed that PHF8 depletion could significantly upregulate the expression of antiviral defense-related genes, whereas *Phf8* reintroduction blocked this effect.

New data

We have supplemented this new data in **Figure S2A** of our revised manuscript. Changes in corresponding result descriptions and figure legends are marked in red.

Question 6: The authors are suggested to add more discussion on the potential mechanism of STEDB1 degradation in the Discussion section.

Response: Thanks so much. We have added several new data and more discussion on the potential mechanism of STEDB1 degradation in the Discussion section of our revised manuscript. Please also refer to our response to **Reviewer #2-Question 3**. All changes in our revised manuscript are marked in red.

Comments of Reviewer #2

In the current study, Liu et al. investigated the role of Phf8 in tumor immunity. It is interesting that Phf8 exerts its function through destabilizing Setdb1, and the authors provide many data in mouse model and cell lines; however, some of the results are not convincing and the mechanistic insight how Phf8 regulates Setdb1 stability is not fully explored. I feel the manuscript should not be published before the concerns are addressed. My detailed comments are as below.

Question 1: In Figure 5E, the majority of H3K9me3 seemed to be repressed after Phf8 KO. The authors claimed Phf8 destabilizes Setdb1. Could the author compare H3K9me3 distribution after Phf8 KO and Setdb1 KO? Whether the two proteins control the same regions on chromatin?

Response: We appreciate the reviewer's insightful questions.

In attempt to clarify the relationship of PHF8 and SETDB1 in regulating genome-wide H3K9me3 levels, we further compared the distribution of H3K9me3 in *Phf8* KO and *Setdb1* KO cells according to the reviewer's suggestion. We have collected publicly available H3K9me3 ChIP-seq data in *Setdb1* KO murine tumor cells (PMID: 33953401; B16) and found 30936 *Setdb1*-dependent H3K9me3 peaks that could be specifically wiped out upon *Setdb1* depletion (please see our new data below, **A**). Our results further showed that approximately 30% (9072/30936) of SETDB1-regulated H3K9me3 signals could be reduced after *Phf8* depletion (**A**, upper right). When noted, H3K9me3 peaks that were co-regulated by SETDB1 and PHF8 mainly located in the transposable element (TE) regions (**B**, upper). We additionally found that H3K9me3 levels of selected retrotransposon loci showed a coincident decrease in *Phf8* KO and *Setdb1* KO cells (**C** and **D**). Collectively, our analysis results indicate that PHF8 and SETDB1 proteins in a large part control the same H3K9me3-marked chromatin regions.

We have supplemented new data A and B in **Figure S6D** and **Figure S6E** of our revised manuscript. Changes in corresponding result descriptions and figure legends are marked in red.

Question 2: Does *Phf8* also bind to the transposons regulated by *Setdb1*? Does *Phf8* KO affect *Setdb1* recruitment to chromatin?

Response: We appreciate these insightful questions.

In our original manuscript, we found that PHF8 barely bound to transposable element regions but mainly bound to gene promoter regions (**A**). This finding was consistent with our analysis results of previously published PHF8 ChIP-seq data (GSE97338; PMID: 28746875) and conclusions of other published studies (PMID: 20421419; PMID: 20622854; PMID: 28100697; PMID: 35179962). During our manuscript revision, we re-analyzed our PHF8 ChIP-seq data and found that PHF8 did not bind to neither PHF8-SETDB1 co-regulated transposons nor SETDB1-regulated transposons (**B**).

To specifically figure out whether PHF8 loss could affect SETDB1 recruitment to chromatin, SETDB1 ChIP-seq analysis in *Phf8* wild-type and KO cells are required. However, we have tried several commercial SETDB1 antibodies available (Proteintech, 11231-1-AP; Genetex, GTX115305) with our best efforts at the time of manuscript revision but did not find a robust antibody for chromatin immunoprecipitation that was qualified for DNA sequencing. In addition, we have analyzed 4 previously published SETDB1 ChIP-seq data in mouse embryonic stem cells (PMID: 35776115; PMID: 19884255; PMID: 19884257; PMID: 26590716) and found SETDB1 binding signals for genomic regions varied a lot among different studies (**C**), suggesting that SETDB1 antibodies used might be ineffective in those assays. We are very sorry that we could not provide answers for this open question at current stage. We would like to make other attempts in the future.

New data

We have supplemented new data B in **Figure S6F** of our revised manuscript. Changes in corresponding result descriptions and figure legends are marked in red.

Related references

1. Nikolaou K, et al. Kmt5a Controls Hepatic Metabolic Pathways by Facilitating RNA Pol II Release from

- Promoter-Proximal Regions. *Cell Rep.* 2017 Jul 25;20(4):909-922. PMID: 28746875
2. Fortschegger K, et al. PHF8 targets histone methylation and RNA polymerase II to activate transcription. *Mol Cell Biol.* 2010 Jul;30(13):3286-98. PMID: 20421419
 3. Liu W, et al. PHF8 mediates histone H4 lysine 20 demethylation events involved in cell cycle progression. *Nature.* 2010 Jul 22;466(7305):508-12. PMID: 20622854
 4. Asensio-Juan E, et al. The histone demethylase PHF8 is a molecular safeguard of the IFN γ response. *Nucleic Acids Res.* 2017 Apr 20;45(7):3800-3811. PMID: 28100697
 5. Moubarak R, et al. The histone demethylase PHF8 regulates TGF β signaling and promotes melanoma metastasis. *Sci Adv.* 2022 Feb 18;8(7):eabi7127. PMID: 35179962
 6. Warriar T, et al. SETDB1 acts as a topological accessory to Cohesin via an H3K9me3-independent, genomic shunt for regulating cell fates. *Nucleic Acids Res.* 2022 Jul 22;50(13):7326-7349. PMID: 35776115
 7. Bilodeau S, et al. SetDB1 contributes to repression of genes encoding developmental regulators and maintenance of ES cell state. *Genes Dev.* 2009 Nov 1;23(21):2484-9. PMID: 19884255
 8. Yuan P, et al. Eset partners with Oct4 to restrict extraembryonic trophoblast lineage potential in embryonic stem cells. *Genes Dev.* 2009 Nov 1;23(21):2507-20. PMID: 19884257
 9. Matsumura Y, et al. H3K4/H3K9me3 Bivalent Chromatin Domains Targeted by Lineage-Specific DNA Methylation Pauses Adipocyte Differentiation. *Mol Cell.* 2015 Nov 19;60(4):584-96. PMID: 26590716

Question 3: Fig. 6D showed that after *phf8* KO, *Setdb1* protein level decreased specific in nuclear in a proteasome-dependent manner. The authors further showed that Phf8 demethylase activity is not required. Could the authors show more mechanisms how Phf8 regulates *Setdb1*? Does Phf8 interact with *Setdb1*? Why does the nuclear fraction specific go down?

Response: We thank the reviewer for these insightful questions that inspire us a lot.

SETDB1 plays a central role in repressive chromatin processes during genome evolution; however, the regulation of SETDB1 expression and stability remain largely unknown to date. To our best knowledge, ATF7IP (also known as MCAF1) is a well-recognized factor that functions as a SETDB1-interacting protein and specifically regulates SETDB1 abundance in the nucleus (PMID: 31576654; PMID: 27732843). Furthermore, ATF7IP-mediated stabilization of SETDB1 is essential for heterochromatin formation. Therefore, we speculated that PHF8 may affect ATF7IP expression or disrupt the interaction of ATF7IP and SETDB1 to control SETDB1 abundance at the time of our manuscript revision. To test this hypothesis, we examined ATF7IP expression and its interaction with SETDB1 in *Phf8* wild-type and knockout cells. Unexpectedly, our results demonstrated that PHF8 loss showed little impacts on neither ATF7IP mRNA expression (please see our new data below, **A**) nor ATF7IP abundance in the nucleus (**B**). Moreover, the interaction of SETDB1 and ATF7IP in the nucleus was not affected by PHF8 ablation either (**C**). These results suggest that ATF7IP was not involved in the process of PHF8-mediated SETDB1 stabilization.

To further explore the underlying mechanisms by which PHF8 regulated SETDB1 stability, we conducted co-immunoprecipitation experiments and found that PHF8 could directly interact with SETDB1 in both CT26 and MC38 cells (**D**). PHF8 is specifically expressed in the nucleus, while

SETDB1 is distributed in the cytoplasm and nucleus; however, whether PHF8 loss recruits other factors to promote SETDB1 proteasomal degradation in the nucleus needs further biochemical testing. We would like to make more attempts in the future.

To clearly present our findings, we proposed a working model illustrating PHF8's function (E). We found that this mechanism of action of PHF8 was quite similar to that of the histone demethylase KDM5B, which also interacted with SETDB1 and regulated SETDB1 stability in the nucleus to silence retrotransposons (PMID: 34671158).

We have supplemented these new data in **Figure S6A-S6C**, **Figure 6F**, and **Figure S6G** of our revised manuscript. Changes in related result description, figure legend and discussion are marked in red.

Related references

1. Tsusaka T, et al. ATF7IP regulates SETDB1 nuclear localization and increases its ubiquitination. *EMBO Rep.* 2019 Dec 5;20(12):e48297. PMID: 31576654
2. Timms R, et al. ATF7IP-Mediated Stabilization of the Histone Methyltransferase SETDB1 Is Essential for Heterochromatin Formation by the HUSH Complex. *Cell Rep.* 2016 Oct 11;17(3):653-659. PMID: 27732843
3. Zhang S, et al. KDM5B promotes immune evasion by recruiting SETDB1 to silence retroelements. *Nature.* 2021 Oct;598(7882):682-687. PMID: 34671158

Question 4: Fig. 6I, *Setdb1* expression greatly repressed the total amount of STING. Does it mean

Setdb1 repress STING expression? At transcription level or at post-translation level?

Response: We appreciate these questions.

At that time of our study, we first examined p-STING^{S365} levels and stripped the same membrane for total STING exposure. The greatly repressed total STING was likely due to incomplete strip of the membrane. To exclude this vague effect, we have prepared new cell lysates and examined p-STING^{S365} and total STING levels in separate gels during our manuscript revision. Our new results showed that ectopic expression of SETDB1 showed little effects on total amount of STING but potently repressed p-STING^{S365} levels (**A**).

Increasing evidence demonstrate that p-TBK1 as a downstream functional readout for antiviral signaling would be more reliable and convincing than p-STING (This point is also raised by **Reviewer #3-Question 1**, expert in STING field). Therefore, we further examined the expression of phosphorylated TBK1 (p-TBK1^{S172}), and our results showed that *Setdb1* overexpression could also abolish PHF8 loss-mediated TBK1 activation (**B**).

We have already supplemented data B in **Figure 6J** of our revised manuscript. Changes in corresponding result description and figure legend are marked in red.

Question 5: Recent publications showed that SETDB1 KO caused cytoplasmic DNA and histone H3. Does it also happen in Phf8-KO cells?

Response: We are thankful for the reviewer's question.

According to recent published work (PMID: 34363353), SETDB1 knockout induces cytoplasmic accumulation of histones H3.1 or cytosolic DNA, in turn activating inflammatory genes. As suggested by the reviewer, we examined cytoplasmic DNA (antibody: Merck, MAB1293, clone AE-2) and histone H3 (antibody: Abcam, ab1791) levels by immunofluorescence analysis in the vector control, *Phf8* KO and *Setdb1* KO cells. Our results showed that PHF8 loss promoted cytoplasmic increase of dsDNA amount and histone H3 puncta formation, phenocopying SETDB1 knockout.

New data

Related reference

1. Wang et al. Epigenetic dysregulation induces translocation of histone H3 into cytoplasm. *Adv Sci (Weinh)*. 2021 Oct;8(19):e2100779. PMID: 34363353

Question 6: In Figure 1M, the effects of two shRNA are not consistent. In IF staining, sh1 caused elevation of IFN- γ , while sh2 did not. In the statistic result, sh1 showed no effect on IFN- γ , which was totally different from IF. Moreover, the staining of Phf8 should be provided here.

Response: Thanks so much for the helpful comments and suggestions.

We are sorry for this inconsistency in our original manuscript. We have carefully corrected this issue by analyzing more fields for each of the five tumor samples per group and selected more representative images for each group in our revised manuscript. As suggested by the reviewer, we additionally performed immunohistochemical analysis for PHF8 expression in sh*Phf8* CT26 tumors (Our PHF8 commercial antibody is not suitable for immunofluorescence assays). Our results show that sh*Phf8* groups displayed a notable decrease in PHF8 abundance as compared with the vector control group.

New data

We have added these new data in **Figure 1M** of our revised manuscript.

Question 7: The changes of several selected REs are not consistent in Figure 4D & 4E.

Response: We appreciate the reviewer’s comment.

To robustly induce the expression of selected retrotransposons, cells were first starved at least 24 hours before IFN- γ stimulation. Long-term starvation probably led to inconsistent changes of retrotransposon expression between *Phf8* wild-type group and KO group when compared with those from normal culture condition. To dispel this confusion, we have decided to remove Figure 4E in our revised manuscript because the data derived from IFN- γ stimulation experiments are dispensable for our study.

Question 8: Based on their TCGA analysis, does PHF8 differently express in tumor tissues? Does its expression correlate with patient survival rates?

Response: We thank the reviewer for these valuable questions. As suggested by the reviewer, we analyzed *PHF8* expression using datasets derived from the TCGA database. We found that *PHF8* was highly expressed in tumors compared with adjacent normal tissues in a variety of cancer types including colorectal adenocarcinoma (COAD) that our study mainly focused on (A). We further examined whether *PHF8* mRNA expression levels in tumors correlated with patient clinical outcomes. Our analysis results demonstrated that *PHF8* overexpression was associated with lower overall survival of COAD or breast cancer patients (B).

We have supplemented colorectal adenocarcinoma data in **Figure 7C** and **Figure 7D** of our revised manuscript. Changes in corresponding result description and figure legend are marked in red.

Question 9: In Sup Fig.1B-D, two sgRNAs were used for Phf8 KO in each cell line, why only one lane for Phf8 KO was shown in Sup Fig. 1A? For other results, why only one KO cell line was selected?

Response: Thanks so much for the helpful questions.

We are sorry that we did not provide enough information for *Phf8* KO clones in our original manuscript. At the time of our study, we prepared several *Phf8* KO clones and selected two of them (*Phf8* sg1 and *Phf8* sg2) for *in vitro* assays. In addition to knockout efficiency examination (**A**), we also tested *Phf8* loss-mediated antiviral responses in these two clones (**B**). Based on the consistency of *Phf8* sg1 and *Phf8* sg2 cells in signaling responses, we therefore selected one of them (*Phf8* sg2) for the rest *in vivo* experiments. Such experimental design with one KO cell clone is referred to several published work (PMID: 34732895; PMID: 33177715).

To exclude the possibility of CRISPR-Cas9 off-target effects, we additionally generated two independent small hairpin RNAs (sh1 and sh2) targeting *Phf8* in CT26 cells (**C**). We consistently observed that *Phf8* knockdown decreased tumor size in immune-competent mice (**D**), substantiating that PHF8 loss induced a potent anti-tumor immunity.

We have supplemented new data A and C in **Figure S1A** and **Figure S1V** of our revised manuscript. Changes in corresponding result descriptions and figure legends are marked in red.

New data

Related references

1. Sun X, et al. Tumour DDR1 promotes collagen fibre alignment to instigate immune exclusion. Nature.

2021 Nov;599(7886):673-678. PMID: 34732895

2. Liu X, et al. Inhibition of PCSK9 potentiates immune checkpoint therapy for cancer. Nature. 2020 Dec;588(7839):693-698. PMID: 33177715

Question 10: Fig. 6E, based on the intensity of DAPI staining, the pictures of Phf8 KO seemed to be less exposed.

Response: We appreciate this helpful comment. We have performed the experiments again and added better-quality images in **Figure 6E** of our revised manuscript.

Question 11: In FigS1B, one of the cell lines seems not proper labelled.

Response: Thanks so much. We are sorry for the mistake. We have corrected this issue in our revised manuscript.

Comments of Reviewer #3

Overall this is a very nice set of experiments. I have several critiques though, which should be addressed.

Question 1: In Figure 2 the pTBK1, pIRF3, and pSTAT1 blots are convincing, but I am not convinced by pSTING – this is a transient species that precedes STING degradation, and if STING were phosphorylated one should see a double band with the total STING antibody. I would remove this piece of data unless the authors can convincingly show the double band with total STING as well. Same with Fig 6I – pTBK1 as a downstream functional readout would be more convincing than pSTING.

Response: Thanks so much for these insightful comments and suggestions.

We agree with the reviewer that p-TBK1 as a downstream functional readout would be much more reliable and convincing. To make our findings clearer, we have removed p-STING and STING immunoblots and replaced them with p-TBK1 and TBK1 immunoblots in our revised manuscript. We consistently found that ectopic expression of *Setdb1* markedly repressed TBK1 activation induced by *Phf8* loss.

We have supplemented this new data in **Figure 6J** of our revised manuscript. Changes in corresponding result descriptions and figure legends are marked in red.

Question 2: For Figure 3A and C – where is the PHF8 blot proving effective KO of PHF8 with each of the other targets? For 3E and F also have they proven effective suppression of all 3 targets? Also, since they were effective individually *in vitro*, why didn't they just test the single KO of STING or MAVS as compared to the double KO *in vivo*?

Response: We appreciate the reviewer's helpful questions.

We are sorry for carelessness in our original manuscript. We have supplemented corresponding immunoblots showing efficient target knockout or knockdown in **Figure 3A**, **Figure 3C**, and **Figure 3E (and F)** in our revised manuscript.

In our study, we discovered that both the RNA-sensing and DNA-sensing pathways were essential for interferon pathway activation induced by PHF8 loss. To validate the critical roles of these two sensing pathways in PHF8 loss-mediated immune activation, we first tried single knockout of STING or MAVS in *Phf8*-deficient cells and found that single knockout of neither of them could not rescue the tumor-forming ability of PHF8 loss *in vivo* even though they were effective individually *in vitro* (data not shown). We deduced that PHF8 depletion exerted anti-tumor immunity so robust *in vivo* that blockade of either single nucleic acid-sensing pathway was hard to abolish PHF8 loss-mediated anti-tumor effects. That's why we simultaneously depleted both of the RNA-sensing and DNA-sensing components for the rescue experiments *in vivo*.

New data

Question 3: Regarding mechanism, prior work from Asensio-Juan et al. (PMID 28100697) has implicated PHF8 in restraining IFN γ gene expression directly involving H4K20 methylation. They authors should reconcile their findings with this additional plausible mechanism.

Response: We appreciate this valuable comment and suggestion.

We have noticed this paper (PMID: 28100697) before we initiated our study. Asensio-Juan and colleagues discovered that PHF8 bound to a subset of IFN γ -responsive gene promoters and kept the promoters in a silent state through the association with HDAC1 and SIN3A. This effect was coupled with low levels of H4K20me1, indicating that PHF8 might function as a transcriptional repressor in response to IFN γ stimulation. Their data were solid; however, this study was conducted *in vitro* and even not in tumor immune models.

In our manuscript, we found that PHF8 inhibition activated retrotransposons and inflammatory genes and subsequently elicited potent anti-tumor immunity. Our work focused on the role of PHF8 in the regulation of retrotransposons upstream rather than IFN γ -responsive genes downstream. Moreover, we validated the role of PHF8 in silencing retrotransposons was regardless of its demethylase activity by carefully testing PHF8 catalytically inactive mutant (*Phf8*^{H247A}) *in vitro* and *in vivo* (Figure 5A-5C in

our manuscript).

In fact, we were curious about this plausible mechanism at that time of our study and discussed with Prof. Liu, who identified PHF8 as an H4K20me1 demethylase in Nature paper (PMID: 20622854). His lab discovered the same findings with ours that PHF8 loss-mediated activation of inflammatory genes and antitumor effect was in a demethylase-independent manner *in vivo*. Collectively, considering the published work (PMID: 28100697) and our study, we deem that PHF8-loss elicits immune responses occur at multiple levels in a demethylase-dependent or -independent manner, which is likely determined by specific biological contexts. We have cited the published work and supplemented more discussion in our manuscript. All changes are marked in red.

Related references

1. Asensio-Juan E, et al. The histone demethylase PHF8 is a molecular safeguard of the IFN γ response. *Nucleic Acids Res.* 2017 Apr 20;45(7):3800-3811. PMID: 28100697
2. Liu et al. PHF8 mediates histone H4 lysine 20 demethylation events involved in cell cycle progression. *Nature.* 2010 Jul 22;466(7305):508-12. PMID: 20622854

Question 4: Finally, simply examining levels of interferon stimulated genes in the 2 human cancer cell lines is not enough. Does PHF8 KO increase pTBK1 and pSTAT1 in these lines? Also, certain human CRC lines have silenced STING (see Xia et al. PMID 26748708). Is the impact of PHF8 KO lost in those human cell lines, or can it de-repress STING, which would potentially be interesting in terms of being able to restore immunogenicity?

Response: We appreciate reviewer's valuable questions.

Following to the reviewer's insightful questions, we additionally detected p-TBK1 and p-STAT1 levels in both HT-29 and LoVo cells during our revision. Our results demonstrated that PHF8 knockout apparently upregulated both p-TBK1 and p-STAT1 in these two cell lines (please see our new data, **A**), consistent with the data from CT26 and MC38 cells in our original manuscript (**Figure 3E**). To figure out whether differential expression of STING in human CRC cell lines would affect PHF8 KO-mediated antiviral immune responses, we further detected total STING protein abundance in our in-house CRC cells (**B**). We found that HT29 cells express relatively higher level of STING, while the expression of STING was relatively lower in LoVo cells. Although HT-29 and LoVo cells showed differential expression of STING, PHF8 loss could still activate the expression of p-TBK1, p-STAT1 and multiple antiviral signaling components in these cells (**Figure 7A and 7B**), indicating that STING abundance seemed not a vital factor that affected PHF8's function. To ascertain this speculation, we further selected another two cell lines, LS180 (STING-high) and DLD-1 (STING-high) cells, and tested p-TBK1 and p-STAT1 levels in their respective *Phf8*-KO cells. Our results consistently showed that PHF8 loss still activated the STING signaling in these cells (**C**). These results collectively suggested that PHF8 loss could de-repress STING even in STING-low CRC cells.

New data

Comments of Reviewer #4

The manuscript is well organized, and the results of intensive experiments are convincing. Furthermore, the results of the analyses using publicly available transcriptome data of human tumors support the hypothesis that the authors proposed. I have some suggestions to improve the manuscript as below.

Question 1: Page 5, line 13. Please clarify the motivation that the authors decided to investigate the functions of PHF8 in anti-tumor immunity in the Introduction section.

Response: We are very thankful for the reviewer's comment.

Accumulating evidence indicate that epigenetic factors are involved in modulating tumor immune microenvironment and regulating antitumor immunity. One of our co-authors, Prof. Wong, identified the role of PHF8 in neuronal differentiation in early years (PMID: 20548336). In recent years, several studies have revealed that PHF8 is aberrantly expressed in malignant tumors and is involved in tumor growth and tumor metastasis; however, the function of PHF8 in anti-tumor immunity has not yet been reported. A recently published work showed an *in vivo* epigenetic CRISPR screen that was conducted to identify cell-intrinsic epigenetic regulators of tumor immunity (PMID: 31744829). When we analyzed the original data of this published work, we found that *Phf8* was also one of candidate genes in the screen and targeting *Phf8*, when lost, potentially enhanced sensitivity to anti-PD-1 treatment in mice. This information prompted us to investigate the role of PHF8 in tumor immune evasion. To make our manuscript clearer, we have added some explanations in the Results section of our revised manuscript. Changes are marked in red.

Related references

1. Qiu J, et al. The X-linked mental retardation gene PHF8 is a histone demethylase involved in neuronal differentiation. *Cell Res.* 2010 Aug;20(8):908-18. PMID: 20548336
2. Li F, et al. In vivo epigenetic CRISPR screen identifies *Asf1a* as an immunotherapeutic target in Kras-mutant lung adenocarcinoma. *Cancer Discov.* 2020 Feb;10(2):270-287. PMID: 31744829.

Question 2: Page 7, line 6. The authors did not present the analysis to assess the synergistic effect between immune checkpoint blockade (ICB) therapy and PHF8 knockdown. The authors should add a statistical analysis specifically to assess the synergistic effect or rephrase the sentence.

Response: We appreciate the reviewer's comment.

We are sorry for inappropriate writing for this drug combination experiment. It is hard for us to calculate combination index showing the synergistic effects of ICB therapy group and *Phf8* knockdown group *in vivo* when they were set with only one dosage. We have rephrased the sentence by removing "synergize with" and replace it with "potentiate" in our revised manuscript.

Question 3: Page 8, line 1. Please add (supplemental) Figures to clarify the correspondence of Fig. 2A and 2C. More specifically, I would like to check whether the up- or downregulated GO terms in CT26 cells are also up- or downregulated in CT26 tumors, respectively (and vice versa).

Response: Thanks so much for the insightful comments and suggestions.

As suggested by the reviewer, we re-analyzed our RNA-seq data during revision. We found that canonical immune signatures, including T cell activation and adaptive immune response pathways were significantly enriched in sh*Phf8* CT26 tumors compared to the control tumors, collectively implying that PHF8 maintains an immune-excluded phenotype.

We have supplemented these data in **Figure S1W** of our revised manuscript. Changes in corresponding result descriptions and figure legends are marked in red.

Question 4: Page 9, line 8. Please clarify why the authors specifically picked up *RLTR46B*, *MER68B*, *LTR67B*, *RLTR13D3*, *MERV1_I-int*, and *L1Md_A*. Please explain the criteria.

Response: We are very thankful for the reviewer's comments.

Our strand-specific RNA-seq analysis of the vector control, *Phf8* KO and *Phf8* KO + *Phf8* CT26 cells revealed that PHF8 could transcriptionally silence retrotransposons. In particular, we identified that a group of retrotransposons including *RLTR46B*, *MER68B*, *LTR67B*, *RLTR13D3*, *MERV1_I-int*, and *L1Md_A* were significantly upregulated ($\text{Log}_2 \text{FD} > 1$, $P < 0.05$) in *Phf8* KO CT26 cells compared with the vector control cells, and importantly, the same set of retrotransposons could also be significantly downregulated ($\text{Log}_2 \text{FD} < -1$, $P < 0.05$) in *Phf8* KO + *Phf8* CT26 cells compared with *Phf8* KO CT26 cells. These results suggested that the transcription of this cluster of retrotransposons was tightly controlled by PHF8. That's why we picked up these retrotransposons and labelled them in **Figure 4A-4C**. We have added some explanations in the Results section of our revised manuscript.

Question 5: Figs. 4A and 4B. The authors should add additional analyses to assess whether retroelements upregulated by PHF8 KO are generally downregulated by the PHF8 supplement.

Response: We appreciate the reviewer's suggestions.

As suggested by the reviewer, we conducted further analysis to show those *Phf8*-regulated retrotransposons. We have supplemented this new data in **Figure S4A** of our revised manuscript. Changes in corresponding result descriptions and figure legends are marked in red.

Question 6: Page 9, line 25. The authors would examine the amount of dsRNA under the RTi treatments.

Response: Thanks. We do not fully understand the reason why we should test dsRAN (Do you mean dsDNA? Typo?) amount after RTi treatment.

In our study, we demonstrated that *Phf8* loss also upregulated the cytosolic DNA-sensing pathway (the cGAS-STING signaling) in addition to the cytosolic dsRNA-sensing pathway, we deduced that the released cytosolic DNA was probably generated through the reverse transcription of retrotransposons (PMID: 31852718). To test this hypothesis, we treated *Phf8* vector control and KO cells with reverse transcriptase inhibitors (RTi) to block dsDNA abundance with retrotransposon origins. And, our results showed RTi treatment led to repress PHF8 loss-triggered dsDNA sensing components (**Figure S4F**) and ISG expression (**Figure S4G**). These results suggest that cytosolic DNA derived from retrotransposon contributed to PHF8 deficiency-mediated antiviral immune responses.

To show the activity of the reverse transcriptase inhibitors (Lamivudine and Nevirapine) in our study, we think that we should examine cytosolic dsDNA amount upon RTi treatments. Our immunofluorescence staining data showed that *Phf8* loss triggered cytosolic dsDNA in the cytoplasm, whereas RTi addition suppressed this effect (Please see our new data below, **A**), indicating that PHF8 loss-induced cytosolic dsDNA increase in a large part originated from retrotransposons through reverse transcription. In addition, we are also curious about the change of cytosolic dsRNA upon RTi treatments. Our results showed that *Phf8* loss-triggered the increase of cytosolic dsRNA was little affected by RTi treatments, which was consistent with unaffected RIG-I and MDA5 expression after RTi treatments (**Figure S4F**).

New data

We have supplemented new data A in **Figure S4E** of our revised manuscript. Changes in corresponding result descriptions and figure legends are marked in red.

Related reference

1. Kwon J, et al. The cytosolic DNA-sensing cGAS–STING pathway in cancer. *Cancer Discov.* 2020 Jan;10(1):26-39. PMID: 31852718

Question 7: Fig.7. To further validate the proposed model using the TCGA dataset, the authors would examine the association between the (total amount of) ERV expression level and the PHF8 expression level in the TCGA (and CCLE) datasets. The expression level of ERVs and genes in the TCGA and CCLE datasets are publicly available here:

<https://data.mendeley.com/datasets/c7r7dw9p42/1>

Response: We appreciate the reviewer’s suggestions. This helps us a lot.

As suggested by the reviewer, we analyzed the correlation of ERV expression and *PHF8* expression levels in the human colorectal adenocarcinoma TCGA cohorts (PMID: 26941318). We divided into *PHF8*-low and *PHF8*-high subgroups according to the mRNA expression levels of *PHF8*. In more

detail, the 25% quantile and 75% quantile of normalized *PHF8* expression were defined as the cutoffs. Tumor samples with normalized *PHF8* gene expression lower than or equal to the 25% quantile were classified into the *PHF8*-low group, while higher than or equal to the 75% quantile were classified into the *PHF8*-high subgroup. We found that many ERVs were enriched in *PHF8*-low subgroup (please see our new data below, **A**). In particular, *LTR9C* and *LTR2B* were anti-correlated with *PHF8* expression (**B**), suggesting that *PHF8* repressed specific ERVs in human colorectal adenocarcinoma. In addition, we also validated these results in a series of colon tumor cell lines using the CCLE database and consistently found that *PHF8* expression was negatively correlated human ERV expression (**C**).

We have supplemented these new data in **Figure S7A-S7C** of our revised manuscript. Changes in corresponding result descriptions and figure legends are marked in red.

Related reference

1. Chuong E, et al. Regulatory evolution of innate immunity through co-option of endogenous retroviruses. *Science*. 2016 Mar 4;351(6277):1083-7. PMID: 26941318

Question 8: Related to Fig. 7. It would be interesting to assess whether the efficacy of ICB therapy is associated with the expression level of *PHF8* in publicly available clinical data.

Response: We appreciate the reviewer's suggestions.

As suggested by the reviewer, we analyzed the *PHF8* expression levels of responders and non-

responders in ICB treatment clinical trials. In both the ICB clinical cohorts of melanoma treated with combined anti-PD-1 and anti-CTLA-4 (PMID: 30753825) and gastric cancer treated with anti-PD-1 (PMID: 30013197), those responders showed significantly lower *PHF8* expression levels than non-responders, suggesting that *PHF8* expression was anti-correlated with ICB response.

We have supplemented these new data in **Figure 7I** of our revised manuscript. Changes in corresponding result description and figure legend are marked in red.

Related reference

1. Gide T, et al. Distinct immune cell populations define response to anti-PD-1 monotherapy and anti-PD-1/anti-CTLA-4 combined therapy. *Cancer Cell*. 2019 Feb 11;35(2):238-255.e6. PMID: 30753825
2. Kim S, et al. Comprehensive molecular characterization of clinical responses to PD-1 inhibition in metastatic gastric cancer. *Nat Med*. 2018 Sep;24(9):1449-1458. PMID: 30013197

Question 9: Page 6, line 20. Please insert a brief explanation why the authors decided to investigate immune memory here.

Response: Thanks so much for this helpful suggestion.

In our study, we observed that *Phf8* knockout potently inhibited tumor formation in immunocompetent mice, and those mice could remain long-term tumor-free (Figure 1E-1H). This promoted us to speculate that PHF8 loss might induce an anti-tumor immune memory. In further experiments, we therefore examined whether those tumor-free mice could resist a rechallenge with parental tumor cells. We have inserted this explanation in our revised manuscript.

Question 10: Page 6, line 27. In order to argue “PHF8 loss promotes the development of an anti-tumor immune memory”, I think that the authors should compare the effect between tumors with and without PHF8. However, such comparison is difficult since the transplantation of the same amount of WT tumor cells is lethal. Therefore, the authors would change the message here.

Response: Thanks so much. We have rephrased the sentence in our revised manuscript.

Question 11: Fig. 4H. I could not understand the result. Maybe the labels KO and KO + Phf8 swapped?

Response: We appreciate the reviewer's suggestions.

We are very sorry that the Y-axis of Figure 4H was not properly labelled in our original manuscript. To clearly present our findings, we have replaced the old data with bar graphs (**A**) in which we showed the FPKMs of TEs, referring to a published paper (PMID: 34253898, Extended Data Figure 2a). Our results demonstrated that several selected retrotransposons exhibited concurrent increase in sense and antisense transcription (bidirectional transcription) upon *Phf8* depletion, which could be decreased by *Phf8* reconstitution (**A**). Our density plots showed an example of a bidirectionally transcribed element (*MERV1_I-int*; **B**)

We have supplemented data A in **Figure 4F** of our revised manuscript. Changes in corresponding result descriptions and figure legend are marked in red.

Related reference

1. Clapes T, et al. Chemotherapy-induced transposable elements activate MDA5 to enhance haematopoietic regeneration. *Nat Cell Biol.* 2021 Jul;23(7):704-717. PMID: 34253898

Question 12: Page 9, line 22. Please add a brief description for the J2 antibody.

Response: Thanks. This has been done in our revised manuscript.

Question 13: Page 14, line 2 and others. Please rephrase “endogenous retrotransposons” as retrotransposons.

Response: This has been done in our revised manuscript.

Question 14: Supplemental Table. Please provide the tables as an excel file instead of a PDF file.

Response: Thanks. This has been done in our revised manuscript.

Reviewers' Comments:

Reviewer #1:

Remarks to the Author:

The authors have made a good effort to address the extensive comments from all reviewers, and I believe the manuscript is almost ready for publication.

Reviewer #2:

Remarks to the Author:

I appreciate the authors' efforts in elucidating the underlying mechanisms. It is great to reveal the interaction between PHF8 and SETDB1. It provides a profound base for SETDB1 regulation. Since the manuscript studies cytosolid DNA regulated by SETDB1, I think the authors should cite the reference they mentioned in the response letter "Wang et al. Epigenetic dysregulation induces translocation of histone H3 into cytoplasm. Adv Sci (Weinh). 2021 oct;8(19):e2100779. PMID: 34363353". I do not have other questions.

Reviewer #3:

Remarks to the Author:

The authors have satisfactorily addressed my concerns

Reviewer #4:

Remarks to the Author:

The authors have adequately addressed our concerns.

We sincerely thank you and the reviewers for the positive comments on our revised manuscript “Loss of PHF8 induces a viral mimicry response by activating endogenous retrotransposons (reference NCOMMS-22-51008A)”. We have address the remaining concerns of the reviewers and editorial requests. Below are the responses to the comments:

Comments of Reviewer #1

The authors have made a good effort to address the extensive comments from all reviewers, and I believe the manuscript is almost ready for publication.

Response: Thanks for the comments.

Comments of Reviewer #2

I appreciate the authors' efforts in elucidating the underlying mechanisms. It is great to reveal the interaction between PHF8 and SETDB1. It provides a profound base for SETDB1 regulation. Since the manuscript studies cytosolid DNA regulated by SETDB1, I think the authors should cite the reference they mentioned in the response letter "Wang et al. Epigenetic dysregulation induces translocation of histone H3 into cytoplasm. Adv Sci (Weinh). 2021 oct;8(19):e2100779. PMID: 34363353". I do not have other questions.

Response: Thanks so much for this suggestion. We have cited this reference in our revised manuscript.

Comments of Reviewer #3

The authors have satisfactorily addressed my concerns.

Response: Thanks for the comments.

Comments of Reviewer #4

The authors have adequately addressed our concerns.

Response: Thanks for the comments.